# SIMWORLD: An Open-ended Simulator for Agents in Physical and Social Worlds

**Xiaokang Ye**[1*]  **Jiawei Ren**[1*]  **Yan Zhuang**[2]  **Xuhong He**[4]  **Yiming Liang**[5]  **Yiqing Yang**[6]

**Xianrui Zhong**[7]  **Mrinaal Dogra**[1]  **Eric Liu**[8]  **Kevin Benavente**[1]  **Rajiv Nagaraju**[8]

**Dhruv Sharma**[1]  **Ziqiao Ma**[9]  **Tianmin Shu**[3†]  **Zhiting Hu**[1†]  **Lianhui Qin**[1†]

[1]UCSD  [2]UVA  [3]JHU  [4]CMU  [5]Purdue  [6]PolyU  [7]UIUC  [8]USC  [9]UMich

[*]Equal contribution.     [†]Equal advising.

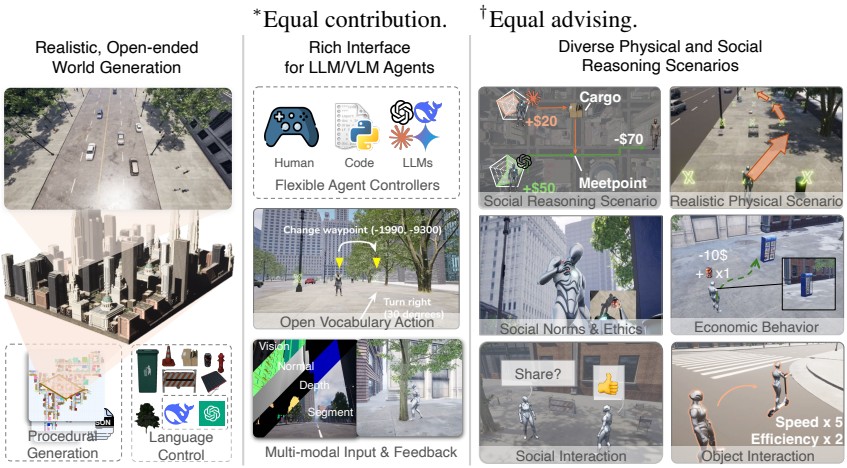

Figure 1: SIMWORLD (1) simulates open-ended environments with realistic physics and social rules, (2) provides native support for LLM/VLM agents to deploy and interact in the simulated worlds, and (3) facilitates the construction of diverse scenarios for AI agent evaluation and development.

## Abstract

While LLM/VLM-powered AI agents have advanced rapidly in math, coding, and computer use, their applications in complex physical and social environments remain challenging. Building agents that can survive and thrive in the real world (e.g., by autonomously earning income) requires massive-scale interaction, reasoning, training, and evaluation across diverse scenarios. However, existing world simulators for such development fall short: they often rely on limited hand-crafted environments, simulate simplified game-like physics and social rules, and lack native support for LLM/VLM agents. We introduce SIMWORLD, a new simulator built on Unreal Engine 5, designed for developing and evaluating LLM/VLM agents in rich, real-world-like settings. SIMWORLD offers three core capabilities: *(1) realistic, open-ended world simulation*, including accurate physical and social dynamics and language-driven procedural environment generation; *(2) rich interface for LLM/VLM agents*, with multi-modal world inputs/feedback and open-vocabulary action outputs at varying levels of abstraction; and *(3) diverse physical and social reasoning scenarios* that are easily customizable by users. We demonstrate SIMWORLD by deploying frontier LLM agents (e.g., `Gemini-2.5-Flash`, `Claude-3.5`, `GPT-4o`, and `DeepSeek-Prover-V2`) on both short-horizon navigation tasks requiring grounded re-planning, and long-horizon multi-agent food-delivery tasks involving strategic cooperation and competition. The results reveal distinct reasoning patterns and limitations across models. We open-source SIM-WORLD and hope it becomes a foundational platform for advancing real-world

agent intelligence across disciplines. *Please refer to the project website for the most up-to-date information:* https://simworld.org.

# 1 Introduction

Large language models (LLMs) have emerged as powerful foundations for intelligent agents, demonstrating strong reasoning capabilities, particularly in domains such as mathematics, coding (e.g., R1, o3), and digital tool use (e.g., web navigation). However, these domains are relatively clean and well-structured, offering clear feedback, unlike the noisy, dynamic nature of the physical and social world where real-world agents are ultimately expected to operate. In practice, such agents must interact with rich, unpredictable environments such as navigating cities, engaging with people, and even making strategic decisions to earn a living [1, 11, 47].

To advance agent development, recent efforts have explored simulation environments that offer rich interactive experiences for training and evaluation (Table 1). However, game-like platforms such as Minecraft [12, 50, 47, 32, 29, 28] and Pokémon [16, 3] provide accessible setups for embodied interaction but lack realistic physical dynamics and social structures, limiting real-world generalization. Domain-specific simulators such as CARLA [10] and AI2-THOR [23] target areas like autonomous driving and household robotics but are limited to narrow task scopes or static environments. Social sandboxes [2, 38] such as Virtual Village [36], simulate interpersonal interactions in scripted, small-scale communities, but lack the open-endedness and scalability required for modeling richer social complexity. Moreover, many of these environments do not support natural language interfaces for goal setting, planning, and control, limiting their compatibility with LLM-based agents.

To meet these growing demands, we present SIMWORLD, a platform designed to support the development and evaluation of LLM (and VLM) agents in complex, dynamic, and interactive environments. SIMWORLD is grounded in three core design principles (Figure 1):

1) **Realistic, Open-Ended World Generation.** SIMWORLD advances simulation by integrating two key capabilities: realism in physical and social dynamics, and open-ended, language-based world generation. On the realism front, SIMWORLD generates hierarchical, city-scale 3D environments grounded in physical laws (e.g., gravity, momentum) and enriched with dynamic features such as lighting, weather, and pedestrian flow. It also embeds socially grounded behaviors—such as obeying traffic signals and maintaining personal space—directly into agent logic to support realistic interactions. On the open-ended side, SIMWORLD supports infinite environment expansion through procedural generation, enabling diverse road networks, building layouts, and urban configurations. Users or AI agents can modify scenes on-the-fly via natural language commands (e.g., "`add a tree next to the hospital`") thanks to SIMWORLD's LLM-based environment editing and asset generation modules, allowing for adaptive environment creation.

2) **Rich Interface for LLM/VLM Agents.** SIMWORLD connects language models with interactive environments by enabling LLM/VLM agents to perceive rich multimodal observations, including visual scenes, abstract layouts, and action feedback. Agents can respond with high-level natural language actions. For example, an agent may reason and generate an abstract action, "`sit on the nearest chair,`" which SIMWORLD translates into a sequence of low-level actions such as moving through waypoints and sitting down. After executing the actions, the simulator provides updated observations and feedback, allowing the agent to adjust its strategy. This closed-loop interaction supports open-ended, language-driven behaviors and empowers agents to perform long-horizon reasoning at a proper abstraction level.

3) **Diverse Physical and Social Reasoning Scenarios.** Building on the above physically/socially grounded environments and LLM/VLM agent interface, SIMWORLD natively supports systematic evaluation and training of agent reasoning in realistic settings. To show how these capabilities work in concert, we present two case studies spanning both physical and social reasoning. *Case Study 1: Grounding Agent Physical Reasoning and Planning* involves a vision-based navigation task where agents interpret visual input, avoid obstacles, and follow traffic rules in real time. *Case Study 2: Strategic Multi-Agent Collaboration and Competition* models an urban delivery economy in which agents bid, invest, and share orders while navigating dynamic environments. With different personas, budgets, and tools (e.g., scooters), agents develop diverse strategies shaped by their goals and shifting conditions (e.g., fluctuating prices). The task highlights complex decision-making and long-horizon planning, where cooperation and competition emerge naturally.

Table 1: **Comparative Analysis of SIMWORLD and Existing Simulators** across key dimensions: **Open-ended World** (procedural scene/asset generation, language-controllable editing), **Physical/-Social Realism** (how close to real-world mechanics) **Action Space** (action abstraction level, open-vocabulary action space), **Agent Type** (types of controllable agents: Humanoid (Hum.), Robot, Drone or Vehicle (Veh.)), and **Physics Engine** (underlying simulation engine). H means high-level actions (e.g., deliver, navigate to), and L means low-level actions (e.g., "forward by 1 step"). See Appendix D for more discussion of related work.

| Simulator | Open-ended World | | Physical/Social Realism | Action Space | | Agent Type | Physics Engine |
|---|---|---|---|---|---|---|---|
| | Procedural | Lang.-Ctrl | | Abstr. | Open-Vocab | | |
| Minedojo [12] | ✓ | ✗ | + | L | ✗ | Hum. | Minecraft |
| Mindcraft [50] | ✓ | ✗ | + | H | ✗ | Hum. | Minecraft |
| MetaUrban [52] | ✓ | ✗ | ++ | L | ✗ | Veh. | PyBullet |
| EmbodiedCity [13] | ✗ | ✗ | +++ | L | ✗ | Drone/Veh. | Unreal Engine |
| CARLA [10] | ✗ | ✗ | +++ | L | ✗ | Veh. | UE & Unity |
| GRUtopia [48] | ✗ | ✗ | ++ | L | ✗ | Hum./Robot | Isaac Sim |
| OmniGibson [25] | ✗ | ✗ | ++ | H/L | ✗ | Robot | Omniverse |
| AI2-THOR [23] | ✓ | ✗ | ++ | L | ✗ | Robot | Unity |
| Habitat 3.0 [39] | ✗ | ✗ | ++ | L | ✗ | Hum./Robot | Bullet |
| VirtualCommunity [64] | ✓ | ✗ | ++ | L | ✗ | Hum./Robot | Genesis |
| UnrealZoo [63] | ✗ | ✗ | +++ | L | ✗ | Hum./Robot/Veh. | Unreal Engine |
| **SIMWORLD** | ✓ | ✓ | +++ | H/L | ✓ | Hum./Robot/Veh. | Unreal Engine |

We evaluate frontier LLMs as agents including `GPT-4o`, `Claude-3.7-Sonnet`, `Gemini-2.5-Pro`, and others on the two challenging tasks described above in SIMWORLD. In the navigation task (Case 1), `GPT-4o` and `Claude-3.7-Sonnet` show strong planning efficiency and high success rates, but consistently ignore red lights and fail to adjust their view to perceive traffic signals. This reveals a disconnect between passive visual perception and active attention, as the models often fail to act appropriately on what they observe. In the multi-agent delivery task (Case 2), agents must operate in a dynamic, competitive economy. `Claude-3.5-Sonnet` and `DeepSeek-V3` earn the highest profits, but often behave erratically—overbidding on low-value orders or spending all their money on scooters they never use. In contrast, `Gemini-2.5-Flash` and `DeepSeek-Prover-V2` follow more conservative, stable strategies, trading peak performance for consistency. Personality traits also shape agent behavior: conscientious agents focus on task completion, while open agents explore but frequently lose money. These findings expose both the strengths and current limitations of LLM-based agents, while revealing rich, often unexpected behaviors that emerge from their interaction with complex environments in SIMWORLD.

## 2 The SIMWORLD Simulator

Realistic, open-ended, and natively LLM/VLM-compatible simulators are essential for developing agents in complex physical and social environments. SIMWORLD takes a step toward this goal with a two-tier architecture: an Unreal Engine (UE) backend and a fully integrated Python layer. The UE backend provides state-of-the-art 3D rendering, real-time physics simulation, and highly interactive, customizable environments. The Python layer bridges this with high-level functionalities, including procedural and language-driven scene generation, rich input-output interfaces for LLM/VLM agents, and modular support for constructing diverse, customizable physical and social reasoning scenarios.

### 2.1 Realistic, Open-Ended World Simulation

**Procedural and LLM-Controllable Environment Generation.** Previous simulators typically rely on a limited set of hand-crafted scenes (e.g., 15 in CARLA [10], 211 in Habitat 3.0 [39]). SIMWORLD develops a procedural generation system [37, Appendix A.1.1] capable of producing diverse, randomized urban environments—including road networks, building layouts, dynamic traffic, and fine-grained elements like street furniture—enabling effectively infinite simulation scenarios. All parameters (e.g., city size, building density, vehicle and pedestrian count) are customizable, allowing users to generate varied and controllable environments with minimal manual effort

Beyond randomized procedural generation, SIMWORLD supports natural language-based scene editing, enabling dynamic world construction through open-ended instructions (Figure 2(a)). Users or AI agents can modify scenes on-the-fly with commands such as "add a red sports car next to the hospital near a museum". SIMWORLD contains a retrieval-augmented LLM-based scene agent that grounds the command by querying the current environment's scene graph. The agent identifies the intended location using spatial anchors (e.g., "hospital") and contextual landmarks

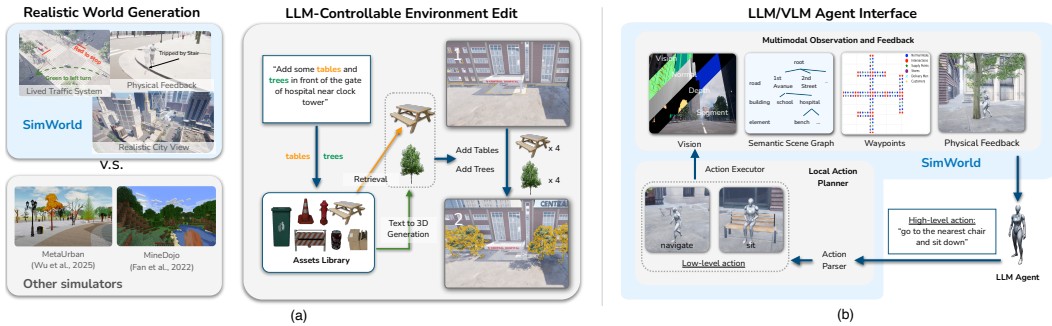

Figure 2: **Realistic Open-ended Simulation and Rich Agent Interface. (a)** SIMWORLD simulates environment with realistic physical and social mechanics (left) and enables language control for real-time environment editing (right) (Section 2.1); **(b)** SIMWORLD supports rich multi-modal inputs and feedback, and accept open-vocabulary high-level actions, to facilitate complex LLM/VLM agent reasoning at proper abstraction levels.

("`museum`"), retrieves a matching asset from a library, and inserts it accordingly. If a suitable asset is unavailable, the agent invokes an off-the-shelf text-to-3D generation model [46] to synthesize a new object from the prompt ("`red sports car`"), converts it into a compatible format, and integrates it into the environment. This approach enables semantically grounded, spatially coherent, and scalable world construction, laying the foundation for interactive and compositional simulation.

**Realistic Physical and Social Simulation.** Powered by UE-5, SIMWORLD provides accurate and continuous physical simulation. Unlike popular environments such as Minecraft, which rely on discrete, block-based mechanics without real gravity or momentum, SIMWORLD models real-world physical dynamics. Agents experience physical forces such as gravity, mass, and inertia: they may slide down slopes, lose balance while pushing heavy objects, or trip over uneven terrain. This realism is essential for supporting grounded, embodied interaction and physically plausible agent behavior.

Social realism is similarly emphasized. SIMWORLD simulates dynamic traffic systems with both vehicles and pedestrians, adhering to real-world norms like crosswalk rules, traffic lights, and personal space. This realism supports high-fidelity, reactive mobility and interaction within urban environments. Together, these physical and social layers provide a rich substrate for evaluating agent reasoning and behavior in real human-like scenarios. Complex multi-agent collaboration and competition dynamics are further supported and described in later sections.

## 2.2 Rich Interface for LLM/VLM Agents

SIMWORLD is designed to make the deployment of LLM/VLM agents straightforward and flexible by exposing a rich, modular input-output interface (Figure 2(b)). This design supports diverse forms of evaluation, reasoning, and training for agents operating in complex environments.

For an agent's inputs, SIMWORLD provides rich multi-modal sensory information and environment feedback that mimic what humans would perceive in the real world. Inputs include raw RGB images, depth maps, segmentation masks, and structured semantic scene graphs, enabling agents to perceive both pixel-level signals and symbolic abstractions. Additionally, agents receive environment feedback in response to their actions. For example, an agent attempting to walk into a closed door would receive a collision event and an updated scene state, allowing it to revise its plan. To further support agent navigation, SIMWORLD implements a waypoint-based abstraction layer, similar as in the popular driving simulator CARLA [10] (Appendix A.1.2). The waypoint system offers high-level reference points along streets and intersections. This enables agents to plan over meaningful spatial landmarks without needing to handle low-level locomotion mechanics. Collectively, these input modalities provide agent developers with a broad design space to tailor LLM/VLM agents based on their desired level of perception and abstraction.

For an agent's outputs, SIMWORLD moves beyond the restrictive, low-level action spaces of many existing simulators (e.g., "move forward by 1 block" in Minecraft). Instead, SIMWORLD supports an open-vocabulary action interface that accepts natural language commands at varying levels of abstraction. For instance, an agent can issue a high-level instruction such as "`go to the nearest chair and sit down`". A built-in **local action planner** interprets this instruction and decomposes

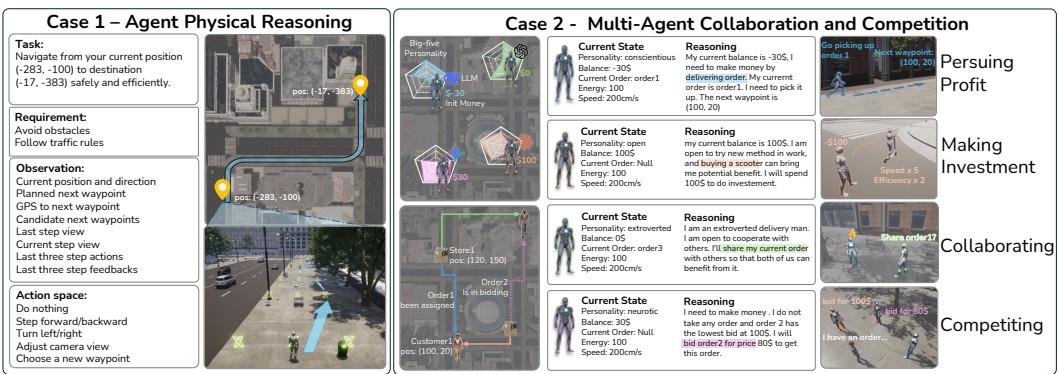

Figure 3: **Overview of Two Case Studies**, showcasing the social-to-physical full stack simulation capability of SIMWORLD. Case 1emphasizes physical reasoning, where agents aim to reach their destinations safely and efficiently. Case 2 focuses on social reasoning, requiring multi-agent collaboration and competition in a real-world delivery scenario. Each agent is initialized with distinct personalities and internal states, and can act to grow, thrive, and ultimately maximize their earnings.

it into a sequence of low-level actions executable by the simulator. The planner consists of two components: an LLM-based parser (e.g., GPT-4o or any user-specified LLMs) that grounds and breaks down high-level intentions (e.g., "`navigate`" and "`sit`"), and an executor that handles actual environment interaction. Depending on the task, the executor can be a simple reliable rule-based system (e.g., following the shortest path to an object) or a VLM-based planner for more complex, perception-sensitive scenarios (e.g., avoiding dynamic pedestrians en route). Our ablation study (Section 3) shows the local action planner substantially facilitates LLM planning.

This layered design enables LLM/VLM agents to focus on long-horizon reasoning and high-level decision-making, while offloading low-level physical execution to the simulator. As a result, SIMWORLD bridges the gap between high-level cognition and low-level embodiment, making it a practical and extensible platform for building and evaluating intelligent agents.

Besides the above advantages, SIMWORLD offers tools for defining tasks, agent roles, rewards, and evaluation metrics. We showcase how to build analyses on SIMWORLD in Sections 3 and 4 (Figure 3), and the workflow for multiagent communication and physical world creation in Appendix A.2.

# 3 Case Study 1: Grounding Agent Reasoning and Planning to Physical World

**Formulation.** Embodied reasoning and physical interaction are evaluated in SIMWORLD through a vision-based urban navigation task. In this task, agents are required to reach a designated goal while circumventing both static (e.g., trees, benches) and dynamic (e.g., pedestrians) obstacles. This necessitates multimodal perception and real-time decision-making under uncertainties and unexpected events [34]. Two distinct settings are considered: *with* and *without* a local action planner. The planner provides a high-level A*-based route without considering obstacles; agents follow this path waypoint-by-waypoint while avoiding obstacles. In the setting without the planner, agents navigate towards the goal position with visual observations. Agents choose from a discrete action set at each step (see below).

**Action Space.** At each decision step, an agent may choose from the following actions: **Do Nothing** (remain stationary), **Step** (move forward or backward for a fixed duration), **Turn** (rotate left or right by a continuous angle), **Change View** (adjust pitch or field of view), and **Change Next Waypoint** (local action planner only, switch to an alternative waypoint).

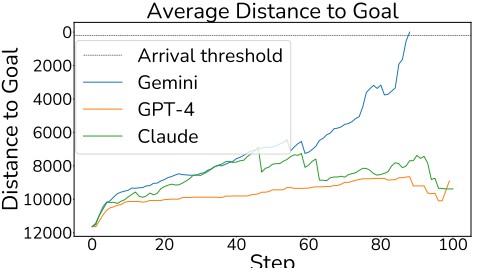

Figure 4: Step-wise average destination distance across tasks for various models.

**Benchmark Environment.** Experiments are conducted with two virtual cities in SIMWORLD, featuring diverse sidewalk layouts and realistic obstacles. Each sidewalk is automatically annotated with both coarse- and fine-grained waypoint graphs to support flexible control and obstacle placement. Static obstacles such as street furniture and trees are randomly placed along fine-grained waypoints, creating

Table 2: **Performance Comparison across Different Difficulty Levels with Static Obstacles.** SR: Success Rate, CC: Collision Count, CC-S: Collision Count (Success Only), RVR: Red-light Violation Rate, STR: Stuck Rate, NDC: Normalized Decision Count. E: Easy, M: Medium, H: Hard. RVR is not 100% as agents may happen to cross during green lights. An ablation study with only red lights presented to the agent resulted in a 100% red-light violation rate, confirming the agent's tendency to ignore red signals when encountered.

| Baseline | SR↑ | | | CC↓ | | | CC-S↓ | | | RVR↓ | STR | | | NDC↓ | | |
|---|---|---|---|---|---|---|---|---|---|---|---|---|---|---|---|---|
| | E | M | H | E | M | H | E | M | H | H | E | M | H | E | M | H |
| Rule-based | 66.73 | 63.33 | 40.0 | 1.80 | 4.14 | 4.00 | 1.55 | 4.74 | 4.67 | 0.00 | 100.00 | 100.00 | 100.00 | N/A | N/A | N/A |
| GPT-4o-mini | 60.00 | 20.00 | 16.67 | 5.97 | 11.13 | 5.33 | 4.50 | 7.33 | 6.00 | 80.00 | 50.00 | 16.77 | 32.00 | 3.98 | 2.28 | 4.25 |
| GPT-4o | 93.33 | 93.33 | **93.33** | 1.73 | 3.37 | 5.10 | 1.64 | 3.46 | 5.07 | 67.86 | 100.00 | 100.00 | 50.00 | 2.22 | 2.48 | 2.32 |
| Claude-3.7-Sonnet | 86.67 | **96.67** | **93.33** | 2.00 | 4.13 | 4.63 | 1.81 | 4.10 | 4.50 | **46.43** | 100.00 | 0.00 | 100.00 | **2.07** | **2.07** | **2.12** |
| Gemini-2.5-Pro | **96.70** | 83.30 | 80.0 | **1.50** | **2.80** | **3.53** | **1.45** | **2.76** | **3.08** | 62.50 | 100.00 | 40.00 | 50.00 | 2.16 | 2.51 | 2.50 |

spatial constraints. Dynamic obstacles (pedestrians) follow predefined motion patterns across selected sidewalks. Tasks are categorized by complexity: **Easy** (a single, straight segment), **Medium** (a single turn at an intersection), **Hard** (multiple turns across four segments), and **Dynamic** (pedestrian interference along a straight path).

**Baseline Agents.** VLM agents (GPT-4o, Claude-3.7-Sonnet, Gemini-2.5-Pro, *etc.*) are compared against a rule-based baseline that strictly follows the planner-generated path. VLM agents are provided with visual input, action history, and task metadata to inform action selection at each step. Further details are available in Appendix B.

**Metrics.** Agent performance is evaluated along three axes: task success, safety, and decision efficiency. We report the following metrics: **Success Rate (SR)**: fraction of tasks completed without timing out, getting stuck, or deviating significantly; **Collision Count (CC)**: total collisions with static or dynamic obstacles; **Collision Count in Success (CC-S)**: collisions in successful episodes, reflecting safety upon goal completion; **Red-light Violation Rate (RVR)**: proportion of successful episodes with red-light violations; **Stuck Rate (STR)**: fraction of failed episodes where the agent remains immobilized for over 2 minutes; **Normalized Decision Count (NDC)**: VLM decisions per waypoint in successful episodes; and **Decision Steps in Success (DSS)**: average number of VLM decisions in successful episodes, indicating planning efficiency under decision constraints. Formal definitions and computation protocols are provided in Appendix B.

## 3.1 Main Results: Agent Planning With Local Action Planner

**Static Obstacles.** We evaluate three levels of task difficulty (Easy, Medium, Hard) under static obstacles with a local action planner. Time limits are set to 15 minutes for Easy and Medium tasks, and 25 minutes for Hard tasks. Four VLMs are evaluated: GPT-4o-mini, GPT-4o, Claude-3.7-Sonnet, and Gemini-2.5-Pro. Results are shown in Table 2, with key insights discussed in Appendix B.7 and summarized below.

> Takeaway: Replanning with Static Obstacles
>
> Model capability plays a decisive role in determining task success, yet current VLMs exhibit limited spatial reasoning and consistently fall short of functioning as truly embodied agents. Despite this, different models display distinct operational strengths, indicating varied but incomplete competencies across the board.

**Dynamic Obstacles.** To assess agents' dynamic obstacle avoidance, we test on tasks with walking pedestrians. VLM agents are limited to 30 decision steps, while the rule-based baseline is given 15 minutes. Results and analysis follow. We summarize the takeaway below and leave experimental details in Appendix B.7.

> Takeaway: Replanning with Dynamic Obstacles
>
> GPT-4o and Claude-3.7-Sonnet excel in dynamic scenarios with high task success and efficient decision-making. Gemini-2.5-Pro shows superior obstacle sensitivity, achieving the fewest collisions, but its cautious behavior leads to more frequent stalls and lower task success (Table 5).

## 3.2 Ablation Study: Without Local Action Planner

To evaluate the robustness of VLM-driven agents under more challenging conditions, we remove the local action planner and re-evaluate their performance on Medium-level tasks. In this setting, agents are only provided with the final destination instead of a pre-defined series of waypoints. They are required to not only avoid collisions but also plan an efficient trajectory autonomously. All agents are restricted to a maximum of 90 decision steps per task. Figure 4 illustrates the progression of the average distance to the goal for each model in a single experimental run. We summarize the key findings below, with detailed experimental settings provided in Appendix B.8.

> **Takeaway: Navigation without Local Action Planner**
>
> `Gemini-2.5-Pro` demonstrates superior autonomous planning and obstacle avoidance without local guidance, while `Claude-3.7-Sonnet` is efficient but less safe. `GPT-4o` fails to adapt without pre-defined waypoints (Table 6).

## 4 Case Study 2: Strategic Multi-Agent Collaboration and Competition

**Formulation.** To evaluate the social reasoning capabilities of foundation models in realistic, open-world urban environments, we propose a strategic multi-agent delivery task that involves both collaboration and competition. LLM-based agents are deployed as delivery personnel in a city-scale environment built using SIMWORLD. Their goal is to grow and thrive in this dynamic setting. To enhance realism and complexity, the environment incorporates several dynamic systems: (1) an energy system, where agents must manage stamina and replenish it through consumables; (2) an economic system, where agents earn and spend currency on purchases such as scooters and drinks; (3) an order-sharing mechanism, enabling agents to collaborate by sharing delivery tasks and optimizing group performance. These components create a rich, interactive simulation environment for evaluating agents' decision-making, adaptability, and social reasoning in complex urban scenarios.

**Environment.** All experiments are conducted on the same map generated by SIMWORLD. To ensure fair and efficient evaluation, graphical rendering is disabled during simulation. However, the physical simulation and evaluation threads remain active to preserve environment dynamics and task fidelity.

**Action Space.** The task features a two-tiered action space: high-level actions decided by LLM-based agents and low-level actions executed by the SIMWORLD local action planner. At each decision step, the agent may choose: **Bid Order** (offer a price to compete for a new order), **Pick Up Order** (navigate to the pick-up point), **Deliver Order** (complete delivery at the destination), **Share Order** (publish the order for collaboration), **Cancel Share Order** (withdraw a previously shared order), **Go to Meet-point** (head to the coordination point for a shared order), **Purchase Drinks** (buy consumables to restore energy), or **Adjust Speed** (cancel order or modify travel speed dynamically). Full action space is shown in Table 7.

**Baseline Agents.** We evaluate multiple foundation models as the backbone of delivery agents, including `Claude-3.5-Sonnet`, `DeepSeek-V3`, `GPT-4o`, `Gemini-2.5-Flash`, and `QWQ`. The ReAct [56] prompting framework is employed to explicitly separate reasoning and action selection. Further implementation details can be found in Appendix C.

**Metrics.** Aligned with the agent's hierarchical decision-making, we design a three-level evaluation framework. Overall performance is measured by **total profit**, while **order success rate**, **energy efficiency**, **order sharing count**, and **investment count** assess operational effectiveness. Additionally, we validate individual actions against simulation rules to ensure stability and evaluate the agent's instruction-following and rule compliance under complex constraints. More details are provided in Appendix C.

### 4.1 Main Results

For each evaluation, we sample 20 agents controlled by the same language model, each running for 5000 simulation steps. At each step, an agent issues two API requests, averaging around 7000 tokens per request. Based on results in Table 3, empirical analysis over three simulation rounds reveals distinct operational behaviors across models. Key insights are discussed in Appendix C.5.1 and summarized below.

Table 3: **Performance of Model-controlled Agents.** Metrics are reported as mean (Avg) and standard deviation (Std) over three 5000-step simulations. Bold indicates the best Avg per column.

| Model | Profit | | Successful Orders | | Energy Efficiency | | Sharing Count | | Investment Count | |
|---|---|---|---|---|---|---|---|---|---|---|
| | Avg | Std | Avg | Std | Avg | Std | Avg | Std | Avg | Std |
| DeepSeek-V3 | **69.48** | 16.77 | 2.10 | 0.47 | 0.34 | 0.07 | 2.33 | 0.47 | 8.00 | 3.00 |
| Claude-3.5-Sonnet | 69.07 | 20.69 | **2.73** | 1.10 | **0.54** | 0.20 | **11.33** | 8.39 | **9.00** | 3.46 |
| GPT-4o | 43.91 | 14.16 | 1.63 | 0.43 | 0.30 | 0.06 | 0.67 | 0.47 | 4.67 | 0.47 |
| Gemini-2.5-Flash | 42.42 | 3.10 | 2.10 | 0.17 | 0.17 | 0.04 | 2.67 | 1.25 | 2.00 | 2.00 |
| Gemini-2.0-Flash | 28.72 | 12.04 | 1.53 | 0.58 | 0.11 | 0.03 | 0.67 | 0.47 | 0.67 | 1.00 |
| Qwen3-32B | 24.73 | 7.95 | 1.37 | 0.13 | 0.40 | 0.17 | 1.33 | 0.47 | 5.33 | 2.06 |
| DeepSeek-Prover-V2 | 21.66 | 7.18 | 0.67 | 0.14 | 0.42 | 0.03 | 7.33 | 8.39 | 1.00 | 1.00 |
| QwQ | 17.31 | 4.07 | 0.87 | 0.20 | 0.41 | 0.20 | 0.33 | 0.47 | 3.33 | 2.52 |
| GPT-4o-mini | 0.00 | 0.00 | 0.00 | 0.00 | 0.00 | 0.00 | 0.00 | 0.00 | 0.00 | 0.00 |

---

Takeaway: Model Performances in Multi-agent Tasks

Top-performing models like `DeepSeek-V3` and `Claude-3.5-Sonnet` achieved high average profits but exhibited significant variability, while `Gemini-2.5-Flash` showed more stable, though moderate, performance. `GPT-4o-mini` failed entirely across all metrics. Overall, the results highlight a trade-off between maximizing average performance and ensuring consistent, reliable behavior.

## 4.2 Ablation Study

To take a step deeper into multi-agent collaboration and competition, we conduct three ablation experiments. **Model Competition**, we sample 24 agents and 12 models that each model controls two agents running the experiment for 1000 rounds. In this ablation experiment, we study how models make choice within highly competitive environment to create as much benefit as they can. **Environment Configuration**, we control two environment configurations, initial financial amount and global order number. For each configuration, we sample several stages from low to high to see how agent's behaviors are influenced by the global environment setting. **Persona**, we use the model with the best performance to control the agents with persona description in the prompts, we sample 20 agents and each persona is allocated to two agents, and we observe how different personas influence the agents' behaviors and decision strategies.

**Model Competition.** To intensify inter-agent competition, we constrain each agent to handle at most one order at a time and set the environment's hunger rate to 0.9, ensuring a high demand for delivery. In each experimental session, 24 agents are jointly controlled by 12 different models, where each model governs two agents. These agents actively bid for orders in a shared environment with the goal of maximizing profit. Each session runs for 1000 simulation steps, and results are averaged across three random seeds.

As shown in Figure 5a, models exhibit distinct bidding behaviors. Notably, `Claude-3.7-Sonnet`, `Gemini-2.5-Flash` and `Gemini-2.0-Flash` demonstrate broad bid price distributions, indicating a flexible bidding strategy. This flexibility increases their chances of winning orders when in competition with other models. In contrast, models such as `LLaMA-4-Scout` and `LLaMA-3.2-11b` tend to use narrower bidding ranges, which limits their competitiveness and results in lower win rates.

Figure 5b presents the head-to-head competition outcomes. `Deepseek-Prover-V2` and `Qwen3-32B` achieve the highest win rates against other models. This is primarily because they often bid lower prices, making their offers more likely to be accepted by the platform. Conversely, models like `GPT-4o` and `LLaMA-3.2-11b` tend to place higher bids, reducing their success rate despite frequent participation. Models such as `QwQ-32B` and `GPT-4o-mini` are less active overall, leading to fewer bids and lower order acquisition rates. This inactivity contributes to their diminished final profit, as shown in Table 3.

Takeaway: Multi-Agent Competition

Models with flexible bidding strategies, like `Claude-3.7-Sonnet` and `Gemini-2.5-Flash`, achieve higher order win rates, while those with narrow or high bids, like `LLaMA-3.2-11b` and `GPT-4o`, underperform. Models that bid aggressively, such as `Deepseek-Prover-V2` and `Qwen3-32B`, dominate head-to-head competitions, whereas inactive models like `GPT-4o-mini` fail to secure bids and profits. `QwQ` and `GPT-4o-mini` show minimal bidding activity and weak task participation (Figure 5).

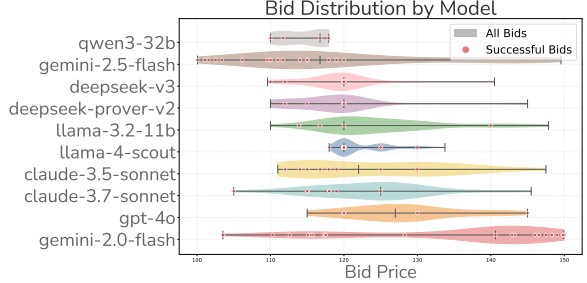

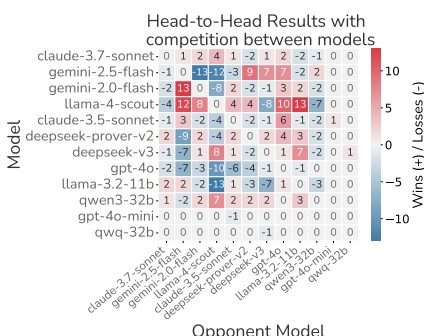

(a) Bid price distribution by model. The violin plots illustrate the distribution of bid prices generated by each model. Red points indicate bids that were successfully accepted.

(b) Win-Loss Matrix of Model Competition.

Figure 5: **Bidding Behavior and Evaluation Results. (a)** Lower bid prices may increase the likelihood of being assigned an order, but often come at the cost of reduced profit margins. **(b)** Higher values in red represent more wins; lower values in blue indicate more frequent losses in pairs.

**Environment Configuration.** We further investigate how different environmental configurations impact agent behavior and overall performance. Specifically, we explore two key factors: the global order availability and the agents' initial financial endowment. For each factor, we conduct a series of controlled experiments to observe how variations affect agents' action distributions.

As shown in Figure 20a (Appendix), when the total number of available orders increases, agents tend to perform fewer pickup and delivery actions and instead choose the `do nothing` action more frequently. This suggests that in resource-rich environments, agents are more inclined to conserve energy and avoid unnecessary effort, opting to wait for optimal opportunities rather than actively pursue deliveries. Conversely, in low-resource settings, agents are more motivated to engage in delivery tasks to secure profits. Additionally, as resource abundance increases, agents demonstrate a higher tendency to initiate and complete shared deliveries, likely as a means to reduce energy costs through collaboration.

Figure 20b (Appendix) illustrates the impact of agents' initial monetary resources. As initial capital increases, agents are less reliant on aggressive bidding and instead prioritize actions such as order pickup. When funds are limited, competition intensifies, leading to more frequent bidding behavior. Furthermore, with sufficient initial capital, agents are more willing to invest in infrastructure, such as purchasing a bike, which enhances their long-term delivery efficiency.

> Takeaway: Resource and Decision-Making Strategy
>
> Order resource scarcity increases agent competitiveness and task urgency. Sufficient agent initial money leads to more relaxed, profit-insensitive behavior (Figure 20 in Appendix).

These observations suggest that agents are more competitive and task-driven in resource-constrained environments. In contrast, resource-rich conditions reduce the urgency to complete tasks and generate immediate profits. Importantly, agents are also more likely to engage in actions that involve upfront costs but promise long-term benefits—such as investment and shared delivery—provided they have the financial capacity and enough taken orders to do so.

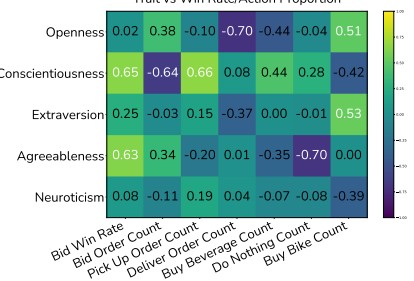

Figure 6: Pearson correlation b/w Big Five personality traits and agent behaviors.

**Influence of Persona.** Personality traits significantly affect the decision-making and performance of delivery agents. As shown in Figure 6, agents with higher Conscientiousness tend to exhibit a lower frequency of bidding actions, a higher frequency of task-completion actions (e.g., picking up orders), and achieve a higher bid win rate. This suggests that conscientious agents prioritize task completion over strategic competition. Agents with higher Agreeableness are less likely to remain inactive (i.e., perform "do nothing" actions) and tend to achieve higher bid win rates. Conversely, agents with lower Agreeableness display higher inactivity and narrower bidding price ranges, limiting their competitiveness. Interestingly, agents with higher Openness exhibit reduced engagement in delivery tasks, possibly because they explore competitive or unconventional bidding strategies that divert attention from task execution.

> **Takeaway: Impact of Persona in Multi-agent Interaction**
>
> Agent personalities shape strategic tendencies: conscientious agents prioritize task fulfillment, while openness and agreeableness modulate competitiveness and inactivity (Figure 6).

## 5    Related Works

Simulations have played a crucial role in constructing environments for training and evaluating autonomous agents. Text-based simulators often emphasize social scenarios, such as human interaction [55], daily activities [36], and relational polarization [38]. Popular embodied simulators support a broader range of applications, particularly in embodied AI research and 2D/3D scene synthesis [27]. However, most embodied simulators remain constrained to either indoor household environments (e.g., AI2-THOR [23], Habitat [39], iGibson [24]) or outdoor driving scenarios (e.g., CARLA [10], MetaDrive [26]) or natural scenes (e.g., AirSim [43]). These simulators [10, 39, 24, 43, 48, 13, 63] often rely on a limited number of manually crafted scenes, which hinders scalability and diversity. Some platforms, such as MetaUrban [52], MetaDrive [26], AI2-THOR [23] and Genesis [4], introduce rule-based procedural generation to alleviate this issue. Nonetheless, existing embodied simulators typically lack support for dynamic multi-agent interactions in complex, outdoor environments.

Recent advancements have introduced large-scale, language-driven social simulators capable of modeling complex societal behaviors. OASIS [55] simulates up to one million LLM-powered agents interacting on social media platforms, capturing phenomena such as information diffusion, echo chambers, and polarization. Casevo [22] integrates chain-of-thought reasoning, retrieval-augmented generation, and customizable memory mechanisms to simulate intricate social phenomena and decision-making processes. MineLand [57] offers a multi-agent Minecraft environment where agents, driven by physiological needs and limited multimodal perception, engage in collective behaviors, fostering ecological and detailed simulations. Project sid [2] further advances this landscape by deploying large number of AI agents within a Minecraft environment to explore the emergence of AI civilizations. VirtualCommunity [64] leverages Genesis to conduct community influence task in out-door, multi-agent scenes. These simulation platforms demonstrate agents' capabilities in forming complex social structures, economies, and governance systems, providing insights into large-scale societal simulations and agentic organizational intelligence.

While prior simulators have demonstrated success in specific domains such as indoor navigation or autonomous driving, none have been explicitly designed to support dynamic, multi-agent interactions in outdoor, city-scale environments with both realistic rendering and physical simulation. SimWorld fills this gap by offering a scalable platform that enables multi-agent collaboration and competition, language-grounded interactions, and procedural generation of urban scenes—all essential for benchmarking advanced embodied agents.

## 6    Conclusion

We presented SIMWORLD, a novel simulator featuring realistic, open-ended world simulation, rich interface for LLM/VLM agents and diverse physical and social reasoning scenarios. We deployed two case studies that highlight key differences in agent performance across models in both single agent physical reasoning and multi-agent social reasoning. These findings underscore the importance of embodied evaluation and open-ended task design in advancing the frontier of AI agent research.

Regarding limitations and future work, although SIMWORLD has demonstrated strong advantages in simulating multi-agent systems within realistic environments rich in physical and visual signals, its deployment for large-scale training, particularly for efficient online reinforcement learning, still leaves room for optimization. We are also excited to build more advanced autonomous agents that can survive and thrive in simulated and real physical worlds, with sophisticated reasoning abilities [14, 17, 2], world models [53, 20], memory [19, 49], emotions, and other capabilities.

We hope that SIMWORLD will serve as a powerful open-source platform for the research community and continue to evolve through collective contributions. Please refer to the project website for the latest updates: https://simworld.org.

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

## A  SimWorld Simulator

### A.1  Python Package

#### A.1.1  Layout Generation

The pipeline of our procedural generation pipe is shown in Figure 7. We have three generation stages which support generating road, building and detail elements in scene graph of city. As Algorithm 1 shows, we use the quad-tree data structure manage the hierarchical structure of scene graph which enables us to support space retrieval function in the language control environment module and support generated different tasks easily.

---

**Algorithm 1:** Layout Generation Algorithm with QuadTree

---

**Input:** Configuration parameters
**Output:** Final merged QuadTree representing the city layout

1 **Function** `RoadMapGeneration`(*config*)**:**
2     Initialize road network grid;
3     Create a new QuadTree for the road map;
4     Place primary roads and secondary roads;
5     Smooth intersections;
6     Insert roads into the QuadTree;
7     **return** *roadQuadTree*

8 **Function** `BuildingGeneration`(*config, roadQuadTree*)**:**
9     Create a new QuadTree for the buildings;
10     Partition land into plots based on road network;
11     Generate building types, shapes, heights;
12     Insert building geometries into the QuadTree;
13     **return** *buildingQuadTree*

14 **Function** `DetailGeneration`(*config, roadQuadTree, buildingQuadTree*)**:**
15     Create a new QuadTree for the details;
16     For each building, add facade details (windows, doors, decorations);
17     For each road, add lanes, crosswalks, signals;
18     Add environmental elements (trees, benches, lights);
19     Insert details into the QuadTree;
20     **return** *detailQuadTree*

21 **Function** `MergeQuadTrees`(*roadQuadTree, buildingQuadTree, detailQuadTree*)**:**
22     Initialize a new QuadTree for the complete city;
23     Merge roadQuadTree into city QuadTree;
24     Merge buildingQuadTree into city QuadTree;
25     Merge detailQuadTree into city QuadTree;
26     **return** *finalQuadTree*

27 **Main Procedure**;
28 roadQuadTree ← `RoadMapGeneration`(*config*);
29 buildingQuadTree ← `BuildingGeneration`(*config, roadQuadTree*);
30 detailQuadTree ← `DetailGeneration`(*config, roadQuadTree, buildingQuadTree*);
31 finalQuadTree ← `MergeQuadTrees`(*roadQuadTree, buildingQuadTree, detailQuadTree*);
32 **return** *finalQuadTree*

---

#### A.1.2  Waypoint System

To facilitate robust agent navigation and path planning within the simulated environment, SIMWORLD incorporates a customizable waypoint system (Figure 8b) that serves as the foundational representation of navigable space. This system is initially constructed using a coarse-grained waypoint skeleton derived from the output of the **Layout Generation** pipeline. Specifically, the pipeline produces high-level geometric features of the environment, such as road centerlines and intersection points, which are then used to define the topological structure of the navigation graph.

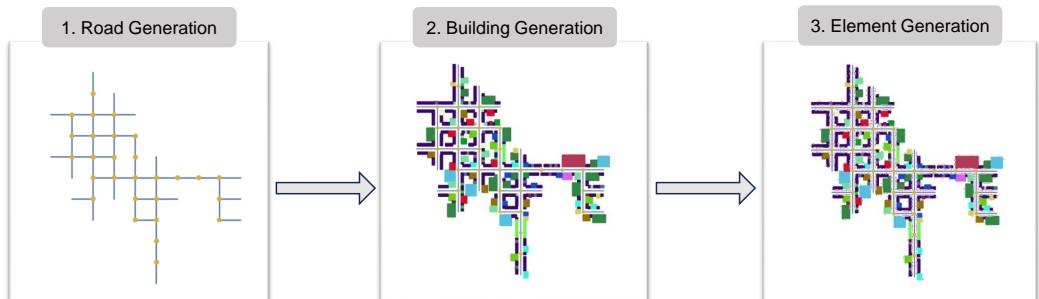

Figure 7: **Generation Pipeline for Static World Initialization.** The process consists of four stages: (1) road network generation, (2) procedural building placement and (3) insertion of street-level elements.

Building upon this base structure, we procedurally generate additional navigable elements, including traffic lanes, sidewalks, and crosswalks. These elements are instantiated by applying spatial offsets relative to the underlying road geometry, enabling the delineation of separate navigation channels for different agent types (e.g., vehicles vs. pedestrians). The result is an initial set of discrete waypoints that capture the essential connectivity of the environment. At this stage, all waypoints are categorized as "intersection" waypoints, reflecting their origin from key structural nodes in the road network.

To improve the granularity and realism of agent motion, we further enrich the navigation graph by interpolating fine-grained waypoints between each pair of adjacent intersection waypoints. These interpolated waypoints are labeled as "normal" waypoints and serve to densify the navigation graph, thereby allowing agents to traverse smoother and more continuous paths. The density of these fine-grained waypoints is configurable through parameters such as interpolation step size and spatial offset magnitude, which can be tuned to suit different simulation scenarios.

This hierarchical waypoint system provides a flexible trade-off between simulation performance and trajectory fidelity. By adjusting the level of waypoint granularity, users can optimize for either computational efficiency or motion realism, depending on the specific demands of downstream tasks such as autonomous navigation, traffic simulation, or human-agent interaction modeling.

### A.1.3 Traffic System

The traffic system is a core component of our simulator, responsible for simulating dynamic road usage by both vehicles and pedestrians. It enables the representation of realistic traffic flow, including vehicle generation, path planning, intersection control, and pedestrian behavior. By managing road interactions and traffic signals, this system supports complex urban scenarios such as congestion, pedestrian crossings, and traffic light coordination, providing a critical foundation for evaluating urban infrastructure and mobility policies.

The traffic system operates on a layout graph provided by the **Layout Generation**, designed to support traffic simulation for any city layout. Based on the coarse-grained waypoints and detailed traffic routes, including traffic lanes, sidewalks, and crosswalks, provided by **Waypoint System**, we sample vehicles, pedestrians and traffic signals in the city. Three managers handle populating the traffic elements:

- **VehicleManager**: Samples positions along the traffic lanes to spawn vehicles, assigning them predefined or dynamically generated routes through the network.
- **PedestrianManager**: Places pedestrians along sidewalks and controls their movement patterns, including crossing decisions at intersections.
- **IntersectionManager**: Identifies intersections within the route network and installs traffic signal agents that control the right-of-way based on configurable timing policies or adaptive logic.

Together, these components form a fully functional traffic simulation pipeline, allowing the virtual city to support realistic and reactive mobility behavior.

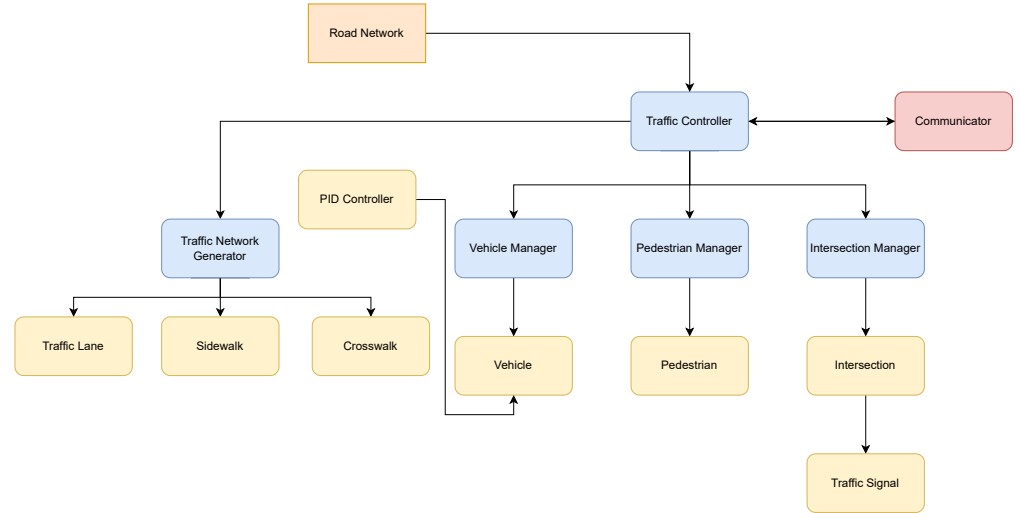

(a) **Architecture of Traffic System.**

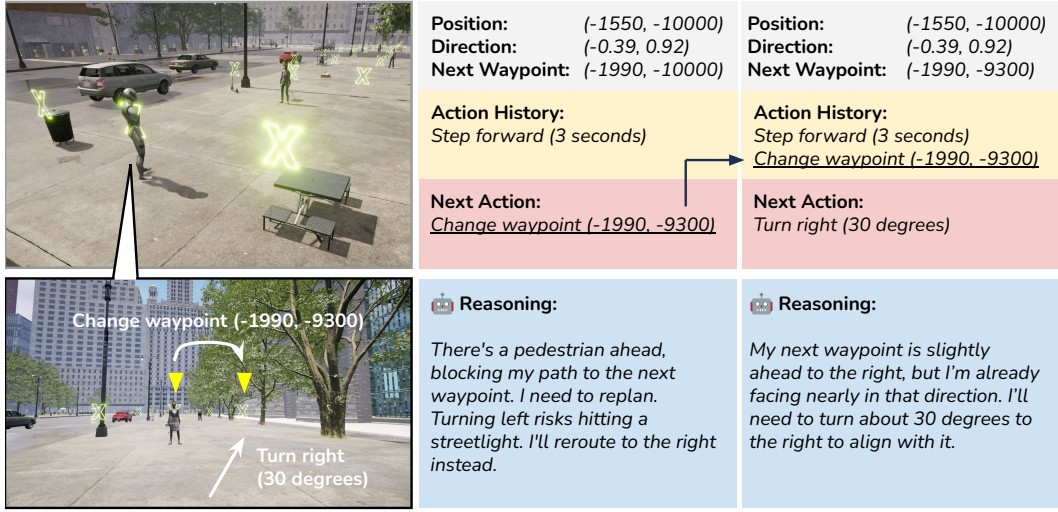

(b) **Waypoint System and Local Action Planner.** The waypoint system provides spatial guidance by specifying the positions and orientations of intermediate navigation subgoals. Based on this information, agents can perform high-level reasoning to plan their behavior. A local action planner operates in conjunction with the waypoint system, generating low-level action suggestions—such as path following or obstacle avoidance—that help agents navigate between waypoints effectively.

Figure 8: **Overview of Traffic and Waypoint Systems.**

As shown in Figure 8a, the traffic system adopts a modular architecture centered around a top-level **TrafficController**, which coordinates the behavior of three specialized managers: **VehicleManager**, **IntersectionManager**, and **PedestrianManager**.

To enable real-time simulation, the system integrates with Unreal Engine through the **Communicator**, which provides bidirectional communication between the traffic logic and the simulation environment. This allows the system to send control signals and receive state updates from virtual actors within the engine.

### A.1.4    Local Action Planner

Recent research on AI agents incorporate action planning modules into their agent's architecture. In our work, we integrate such a module into the simulator itself, providing a useful interface for LLM agents to use and assist the agents' high-level long-horizon reasoning. LLM-Planner [44]

uses visual information of the environment for decomposing high level plan into low level actions. CoELA [58] uses a rule-based only executor for execution. Compare to their similar module, ours are more comprehensive in that we include rule-based and LLM as executor; we receive both information in the form of natural language as well as in the form of visual; our local action planner module also extendable to corresponding with our extendable simulator. Our local action planner module is devided into two part, parser and executor. Parser simply receive the high-level plan from user LLM and parse the high-level plan into low-level executable actions, while executor is responsible of executing the parsed low-level actions one by one given the environmnet information in the simulator. Considering the variety of needs from different researchers, we implement local action planner executor module using two different ways: rule-based and visual based. For perceiving the environment of the simulator: rule-based local action planner will receive abstract city layout information, and visual-based local action planner will capture the visual information from the environment directly, which allows the module to be adapted to different visual language models and further improve the comprehensiveness. For non-atomic action like navigation, we integrate a route planner based on the Dijkstra's algorithm[9] for rule-based local action planner, which provides ground-truth solutions for navigation tasks. This route planner serves as a supporting tool for the local action planner module, ensuring that natural language descriptions of activities can be reliably converted into executable low-level action sequences. For visual-based local action planner on the other hand, we simply feed the visual information to backbone visual language model and let the model decide what to do next to test the capability of visual language model.

For example, when local action planner receives a user plan such as "go to the nearest chair and sit down," it decomposes the instruction into an action list: *navigate*, *agent_sit_down*. For the *navigate* action, the rule-based mode first computes the shortest path from the agent's current position to the nearest chair, which results in a sequence of navigation actions such as $navigate(0, 1)$, $navigate(1, 10)$, and $navigate(10, 10)$, where $(10, 10)$ is the location of the chair. Once the agent reaches the chair, local action planner proceeds to the next action, $agent\_sit\_down$, and completes execution as the action list becomes empty.

### A.1.5 LLM-based Environment Editing

SIMWORLD supports real-time, language-driven environment editing, enabling the creation of open-ended, generative worlds through natural language. Users interact via a Python interface, issuing instructions such as "add a tree next to the hospital" or "place a red sports car near the fountain." To handle such commands, we introduce a dual-stage pipeline: (1) asset retrieval and placement, and (2) text-to-3D asset generation.

For asset retrieval and placement, given an instruction like "Add a bench near the cafe; I can see a museum and a skyscraper nearby," an LLM first parses the target asset, spatial anchor (e.g., "the cafe"), and contextual landmarks ("museum" and "skyscraper"). Candidate reference assets are located via semantic search on the scene graph and ranked by comparing the described surroundings with actual neighbors using embedding-based similarity. The top-ranked reference point determines the target location. The desired asset is then retrieved from the library using CLIP-based[42] matching and placed accordingly.

If no suitable asset is found in the database, we fall back to a text-to-3D generation module. Given a prompt such as "a red sports car," the system generates a new asset using a generative model, converts it into a usable format, and integrates it into the scene in real time.

This approach allows for flexible, scalable environment editing that is both semantically grounded and spatially coherent—laying the foundation for open-ended simulation and interactive world modeling.

### A.2 Communicator and Seneraios Creation

### A.2.1 Communicator

Inspiring by UnrealCV[41], the Communicator module serves as the bridge between Python-based simulation logic and Unreal Engine's real-time 3D environment. Implemented in both Python and C++, it uses UnrealCV to establish a TCP connection, enabling asynchronous communication between the two sides.

We extended the basic communication protocol by defining a custom set of commands for scene control, actor manipulation, and data querying. These commands are encoded in JSON format and passed through the TCP channel. For instance, Python can instruct UE to spawn an actor at a specific location, retrieve the current pose of a pedestrian, or trigger motion events on a robot.

The Communicator plays a central role in the simulation loop: Python uses it to update the virtual world, while UE reports back environmental feedback or visual state. This design allows us to decouple logic computation from rendering, achieving both flexibility and modularity.

### A.2.2  Task Suite Workflow

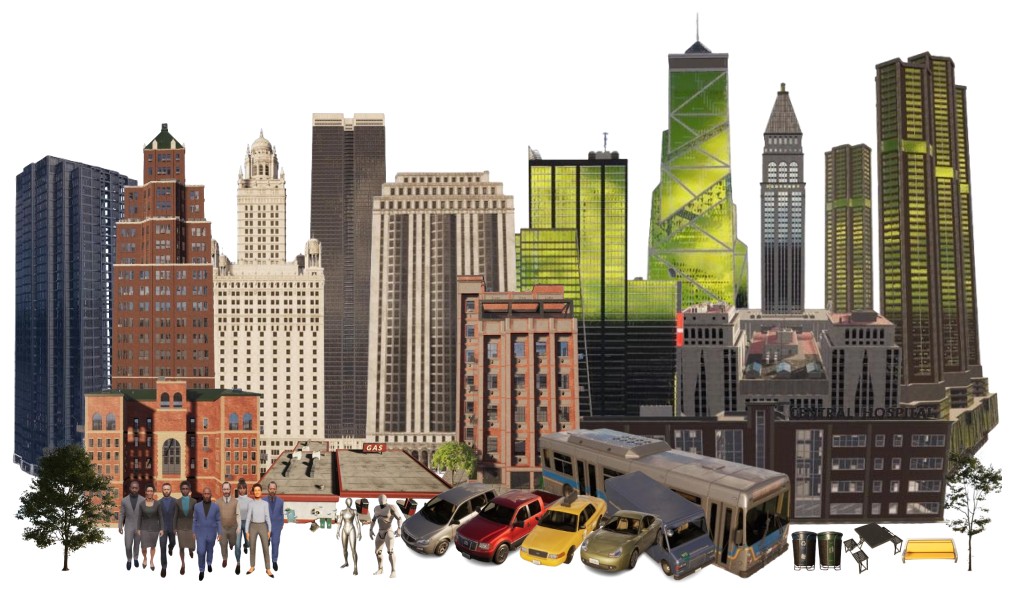

(a) **A Subset of Collected Assets.**

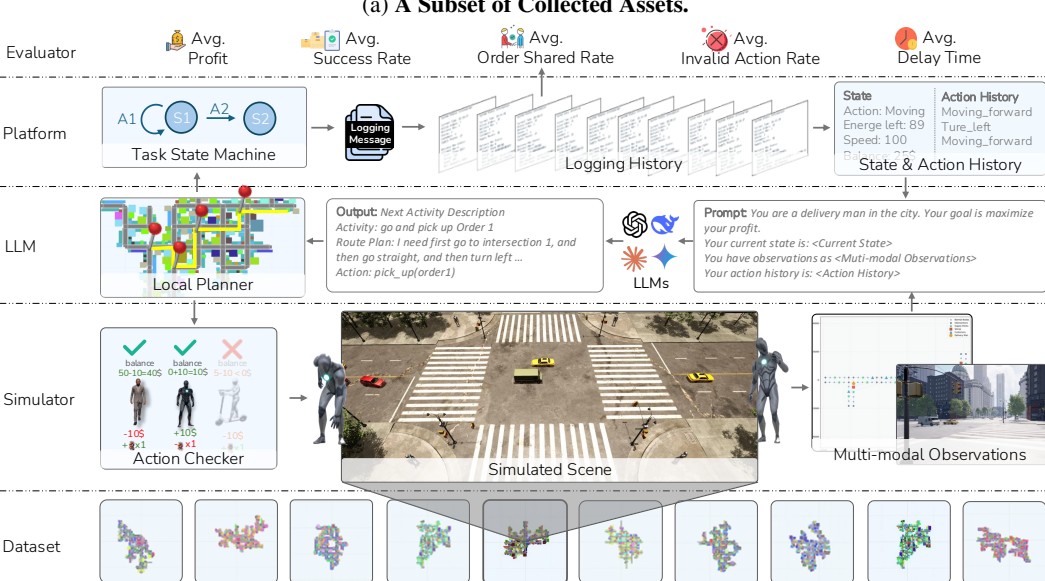

(b) **Workflow for Constructing a Task Suite.** We implement a delivery task as a representative example within the SIMWORLD simulator. The full task suite includes five key components: (1) Dataset, (2) Simulator, (3) LLM, (4) Platform, and (5) Evaluator.

Figure 9: **SIMWORLD Assets and Task Suite Workflow.**

| Action | Category | Description |
|---|---|---|
| Move Forward | Navigation | Keep moving in the current direction |
| Step Forward/Backward | Navigation | Step forward/backward for a fixed time |
| Rotate | Navigation | Turn the body to face a new direction |
| Stop | Navigation | Stop moving |
| Look Up/Down | Observation | Adjust the gaze upward/downward by a degree |
| Focus | Observation | Adjust the field of view |
| Pick Up | Object Interaction | Grasp and lift an object |
| Drop Off | Object Interaction | Release a held object at the target location |
| Sit Down | Object Interaction | Transition to a seated position |
| Stand Up | Object Interaction | Rise from a seated position |
| Open Door | Object Interaction | Interact to open a door |
| Enter Car | Object Interaction | Get into a vehicle |
| Exit Car | Object Interaction | Leave a vehicle |
| Carry Heavy Object | Object Interaction | Transport a heavy object |
| Put Down Heavy Object | Object Interaction | Place a heavy object on the ground |
| Ride Scooter | Object Interaction | Control and ride a scooter |
| Have Conversation | Social Action | Exchange verbal communication |
| Point Direction | Social Action | Gesture to indicate direction |
| Wave Hand | Social Action | Signal or greet with a hand wave |
| Discuss | Social Action | Engage in dialogue or explanation |
| Argue with Body Language | Social Action | Express disagreement using gestures |
| Dance | Social Action | Perform rhythmic movement with the body |
| Make Phone Call | Social Action | Simulate or engage in a phone conversation |

Table 4: **Comprehensive List of Actions with Categories and Descriptions.**

## A.3 Unreal Engine Integration

### A.3.1 Action Space

Table 4 presents the complete set of low-level actions supported by SIMWORLD.

### A.3.2 Assets

Our simulator provides a rich collection of city-scale assets, designed to support realistic and diverse urban simulations. These assets include buildings, trees, street furniture, vehicles, pedestrians, and robots (Figure 9a). All assets are sourced from the Unreal Engine Marketplace to ensure high visual fidelity and performance.

Below is a selection of the assets currently available in our simulator:

- **Buildings**: A variety of architectural styles, including residential, commercial, and industrial structures.
- **Trees**: Multiple tree species with seasonal variations to enhance environmental realism.
- **Street Furniture**: Items such as benches, streetlights, mailboxes, and trash bins to add detail and immersion.
- **Vehicles**: A range of vehicles including cars, buses, trucks, and bicycles, each with accurate scale and animations.
- **Pedestrians**: Human characters with diverse appearances and animations to simulate crowd behavior.
- **Robots**: Humanoid robots for testing autonomous navigation and interaction.

These assets collectively enable the creation of complex, dynamic, and realistic city scenes for simulation, visualization, and research purposes.

In addition to the curated asset library, we provide an **Asset Generation Pipeline** that leverages Text-to-3D models[46] to generate 3D assets directly from natural language descriptions. This tool streamlines the content creation process by translating user prompts into usable Unreal Engine assets, thereby significantly lowering the barrier to customizing city environments.

# B  Case Study 1 Details

## B.1  Formulation

To evaluate the embodied reasoning and physical interaction capabilities of agents within SIMWORLD, we introduce a vision-based navigation task in urban environments. Agents are tasked with reaching a specified destination while avoiding both static obstacles (e.g., trees, benches) and dynamic agents (e.g., pedestrians), necessitating strong spatial reasoning and multimodal perception under uncertainty. The task requires agents to process real-time visual input and make sequential decisions to avoid collisions, respect traffic rules, and reach the goal efficiently.

To analyze the interplay between high-level planning and low-level reactive control, we consider two settings: *with* and *without* a Local Action Planner. In the Local Action Planner setting, agents are guided by a precomputed global route generated via A* search, which omits obstacle information. Agents must navigate from one waypoint to the next along this route while handling local interactions. In the planner-free setting, agents receive only the goal position and are expected to autonomously plan and execute a safe trajectory.

## B.2  Action Space

Each agent may choose from the following actions at each decision step:

- **Do Nothing**: Remain stationary.
- **Step**: Move forward or backward for a fixed duration.
- **Turn**: Rotate left or right by any angle (continuous range).
- **Change Next Waypoint**: (Local Action Planner only) Switch to a different candidate waypoint.

## B.3  City Map and Task Generation

- **Coarse-grained Map**: Each sidewalk is divided into three waypoints (start, midpoint, end), forming a navigation graph to generate tasks.
- **Fine-grained Map**: Each sidewalk is sampled with ∼70 evenly spaced waypoints. These serve both as candidate motion targets and obstacle anchor points.
- **Obstacles**: Static objects (trees, trash bins, etc.) are randomly placed on fine-grained waypoints. Dynamic pedestrians follow sidewalk paths at moderate speed with randomized starting positions.

Figure 10 illustrates the task generation pipeline. We begin with a basic road network provided by SimWorld, which includes sidewalks, crosswalks, and intersection nodes. To construct the coarse-grained navigation graph, we interpolate a midpoint for each sidewalk segment, yielding a simplified map structure suitable for high-level planning.

Tasks are generated by randomly sampling start and goal nodes from this graph, with constraints on the number of segments between them to control difficulty. For fine-grained navigation, we densify the original sidewalk geometry by interpolating evenly spaced waypoints. Each sidewalk is divided into four lateral lanes, each containing 17 waypoints, while each crosswalk contains 8 waypoints.

To simulate realistic urban clutter, we place static obstacles such as trees and street furniture along these waypoints. Trees are sampled on the innermost lane (nearest to buildings) of each sidewalk, while other street elements (e.g., benches, trash bins) are distributed across the remaining three lanes. Importantly, for each group of four laterally aligned waypoints (i.e., across the four sidewalk lanes at the same longitudinal position), at least one is guaranteed to remain obstacle-free. This ensures that there is always a traversable option at every step along the sidewalk.

We sample 40 tasks spanning the four difficulty levels:

- Easy (10 tasks), Medium (10 tasks), Hard (10 tasks), Dynamic (10 tasks).
- Each task is defined by a sequence of waypoints $T_i = [W_1, W_2, ..., W_n]$.

## B.4  Baseline Agent Protocols

- **Rule-based Agent**: Follows Local Action Planner waypoints using a fixed stepping strategy. Obey the traffic rules and does not observe visual input or adapt to obstacles.

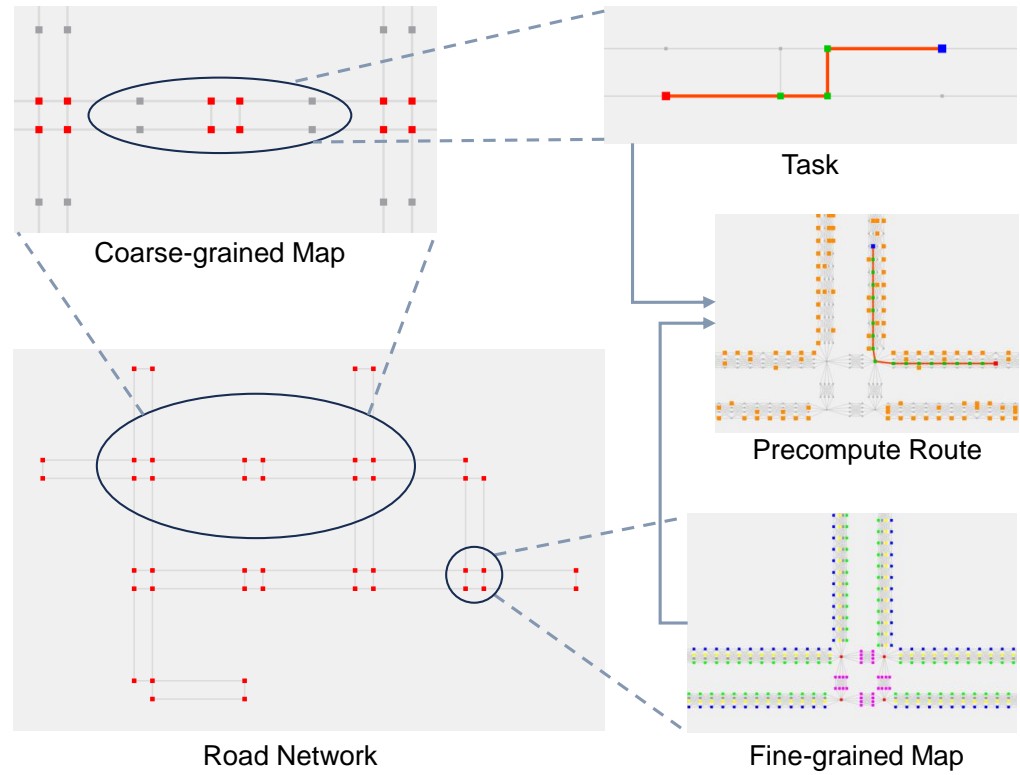

Figure 10: **Task Generation Pipeline for Case Study 1.**

- **MLLM Agents**:
    - Input: current image, last-step image, past 3 actions and feedback, current navigation goal, relative distance and angle to current navigation goal.
    - Output: action selected from the predefined space.
    - Perception is simulated via RGB screenshots with 90° horizontal FOV.

### B.5 Metrics

Throughout the experiment, we evaluate agent performance using the following six metrics, which together capture task success, safety compliance, and decision efficiency:

- **Success Rate (SR)**: The percentage of navigation tasks in which the agent successfully reaches its destination without getting stuck, timing out, or deviating significantly from the designated route. This metric reflects the overall effectiveness of the agent's decision-making and path-following capabilities.

$$\text{SR} = \frac{N_{\text{success}}}{N_{\text{total}}}$$

where $N_{\text{success}}$ is the number of successfully completed tasks, and $N_{\text{total}}$ is the total number of tasks.

- **Collision Count (CC)**: The total number of collisions that occur between the agent and either static obstacles (e.g., trees, trash bins) or dynamic entities (e.g., pedestrians). A lower collision count indicates better obstacle avoidance and situational awareness. Collisions are recorded for all episodes, regardless of success or failure.

$$\text{CC} = \sum_{i=1}^{N_{\text{total}}} c_i$$

where $c_i$ is the number of collisions in episode $i$.

- **Collision Count (Success Only) (CC-S)**: The number of collisions occurring only within successful episodes, which reflects if the agent really avoids the obstacles. Since the CC is possibly low if the agent get stuck at the very beginning. This metric helps assess how safely the agent completes tasks when it does reach the goal.

$$\text{CC-S} = \sum_{i \in \mathcal{S}} c_i$$

where $\mathcal{S}$ is the set of successful episode indices.
- **Red-light Violation Rate (RVR)**: The fraction of successful episodes in which the agent crosses an intersection during a red traffic signal, violating traffic rules. This metric evaluates the agent's ability to obey traffic signals and is computed only over successful episodes to ensure fair assessment under comparable task completion.

$$\text{RVR} = \frac{\sum_{i \in \mathcal{S}} r_i}{|\mathcal{S}|}$$

where $r_i = 1$ if a red-light violation occurred in episode $i$ and 0 otherwise, and $|S|$ is the total number of successful episodes.
- **Stuck Rate (STR)**: The percentage of failed episodes in which the agent remains unable to reach the destination due to either physical immobilization (e.g., blocked by obstacles) or ineffective decision-making (e.g., repeatedly turning or repositioning without forward movement) over a period of 2 minutes. STR captures both hard failures (being physically stuck) and soft failures (behavioral indecision), reflecting challenges in planning and low-level control.

$$\text{STR} = \frac{N_{\text{stuck}}}{N_{\text{failed}}}$$

where $N_{\text{failed}} = N_{\text{total}} - N_{\text{success}}$ and $N_{\text{stuck}}$ is the number of failed episodes classified as stuck.
- **Normalized Decision Count (NDC)**: The ratio between the total number of decision-making by the large model and the total number of fine-grained waypoints provided by the Local Action Planner, computed over successful episodes. NDC measures the agent's decision efficiency, with a lower value indicating more economical use of the model.

$$\text{NDC} = \frac{1}{|\mathcal{S}|} \sum_{i \in \mathcal{S}} \frac{d_i}{w_i}$$

where $d_i$ is the number of model decisions made in episode $i$, and $w_i$ is the number of fine-grained waypoints along the successful path. $|S|$ is the total number of successful episodes.
- **Decision Steps in Success (DSS)**: The average number of decision-making steps taken by the large model in episodes that were successfully completed. DSS measures the typical number of decisions required to achieve a successful outcome, with a lower value potentially indicating more efficient or direct decision-making in successful scenarios.

$$\text{DSS} = \frac{1}{|\mathcal{S}|} \sum_{i \in \mathcal{S}} d_i$$

where $d_i$ is the number of model decisions made in successful episode $i$, and $|S|$ is the total number of successful episodes.

Together, these metrics provide a comprehensive evaluation of an agent's navigation performance, safety behavior, and decision-making efficiency in complex urban environments.

## B.6 Prompt Design

### B.6.1 With Local Action Planner

---

**System Prompt of Navigation Task with Local Action Planner**

```
SYSTEM_PROMPT = """
You are an embodied agent in a 3D simulation environment, where the unit is
    centimeters (cm). Your task is to navigate from your current position to a
    specified destination safely and efficiently. A path to the destination is
    provided as a list of waypoints. You only need to go to the next waypoint
```

---

```
        if it's not blocked. Otherwise, you should choose a new next waypoint. For
        this task, please only output a parsable json string inside brackets.
        Please start your answer with { and end your answer with }. Don't include
        any notes or explanations with the output json string.
    """
```

Figure 11: **Examples System Prompt of Navigation Task with Local Action Planner.**

---

**User Prompt of Navigation Task with Local Action Planner**

```
USER_PROMPT = """
Currently you are at {current_position} and your direction is {current_direction
    }. Your final destination is {target_position}. Your next waypoint is {
    next_waypoint}, among the possible next waypoints: {possible_next_waypoints
    }. The next waypoint is {relative_distance:.2f} cm away from you. The
    relative angle to the next waypoint is {relative_angle:.2f} degrees (
    negative means next waypoint is to your left, positive means next waypoint
    is to your right). Your walking speed is 200 cm/s.

You are given two images:
- Previous view (1 step ago): shows what you saw before your last decision.
- Current view: shows what you see now.
Use the two images to understand the changes in your surroundings and help you
    make a better decision.

You have the following action history (most recent at the bottom):
{action_history}
Before making your decision, you should consider the history of actions. Avoid
    choosing a new waypoint multiple times in a row. If your last action was
    choosing a new waypoint, you should now try moving toward it unless it's
    truly blocked. Avoid turning around multiple times in a row. You should try
     to move forward a little bit before turning around.

You must:
- Avoid obstacles and pedestrians on the street.
- Obey traffic lights when at intersections: stop at red lights, proceed on
    green.
- Use both current and previous visual observations to assess your surroundings.
- Only adjust your walking direction when it deviates more than 15 degrees from
    the next waypoint.
- Avoid selecting a new waypoint repeatedly or turning around too frequently.
- Never walk forward continuously for more than 5 seconds; 1-5 seconds is
    acceptable.

Think step by step and reason about your decision.
- If you are facing your next waypoint and it's not blocked, you should move
    forward for a short time (1-5 seconds).
- If you are not facing your next waypoint roughly, you should turn around to
    face the next waypoint.
- If your next waypoint is blocked, you should choose a new waypoint from the
    possible next waypoints.
- If you get stuck, you can turn around and change your next waypoint or step
    backward for a short time (1-5 seconds).
- If you are at an intersection, you should stop and wait for the pedestrian
    light to turn green before moving.
- If you need to change your view, you can look up or set your field of view.

Now it's time for you to make a decision. You have the following options:
- 0: Do nothing.
- 1: Step. Must specify duration (max 5 sec) and direction (0 = forward, 1 =
    backward).
- 2: Turn. Specify angle (0-180 degrees) and direction (clockwise = true/false).
- 3: Look up. Specify look_up_angle (-90 to 90 degrees).
- 4: Set field of view. Specify fov (0-180 degrees).
- 5: Choose a new waypoint. Must specify one from the possible next waypoints.

Output a single json object string, whose keys are 'choice', 'duration', '
    direction', 'angle', 'clockwise', 'look_up_angle', 'fov', 'new_waypoint'.
"""
```

Figure 12: **Examples User Prompt of Navigation Task with Local Action Planner.**

### B.6.2 Without Local Action Planner

---

**System Prompt of Navigation Task without Local Action Planner**

```
SYSTEM_PROMPT = """
You are an embodied agent in a 3D simulation environment, where the unit is
    centimeters (cm). You are good at navigating in a city environment. Your
    task is to navigate from your current position to a specified destination
    safely and efficiently. For this task, please only output a parsable json
    string inside brackets. Please start your answer with { and end your answer
     with }. Don't include any notes or explanations with the output json
    string.
"""
```

Figure 13: **Examples System Prompt of Navigation Task without Local Action Planner.**

---

**User Prompt of Navigation Task without Local Action Planner**

```
USER_PROMPT = """
You are currently at {current_position} and your direction is {current_direction
    }. Your final destination is {target_position}. The destination is
    approximately {relative_distance:.2f} cm away, and the relative angle to it
     is {relative_angle:.2f} degrees (negative = to your left, positive = to
    your right). Your walking speed is 200 cm/s.

You are given two images:
- Previous view (1 step ago): what you saw before your last decision.
- Current view: what you see now.
Use these images to understand how your environment is changing and make smart
    decisions.

You have the following action history (most recent at the bottom):
{action_history}
Before making your decision, you should consider the history of actions.

The relative distance and angle only give you a rough idea of where the
    destination is. You must not walk directly toward the destination without
    checking the surroundings. Carefully plan your path instead.

You must:
- Avoid obstacles on the sidewalk.
- Keep walking on the sidewalk. Do not step into the roadway, crash into
    buildings, or hit obstacles on your way.
- Use both current and previous visual observations to assess your surroundings.
- Never walk forward continuously for more than 5 seconds; 1-5 seconds is
    acceptable.

You are required to make a decision for your next action. You have the following
     options:
- 0: Do nothing.
- 1: Step. Must specify duration (max 5 sec) and direction (0 = forward, 1 =
    backward).
- 2: Turn. Specify angle (0-180 degrees) and direction (clockwise = true/false).

Output a single json object string, whose keys are 'choice', 'duration', '
    direction', 'angle', 'clockwise', 'new_waypoint'.
"""
```

Figure 14: **Examples User Prompt of Navigation Task without Local Action Planner.**

### B.7 Main Results: Agent Planning With Local Action Planner

**Static Obstacle.** We evaluate three levels of task difficulty (Easy, Medium, Hard) under static obstacles with a local action planner. Time limits are set to 15 minutes for Easy and Medium tasks, and 25 minutes for Hard tasks. Four MLLMs are evaluated: GPT-4o-mini, GPT-4o, Claude-3.7-Sonnet, and Gemini-2.5-Pro (gemini-2.5-pro-preview-03-26). Results are shown in Table 2, with key insights discussed below:

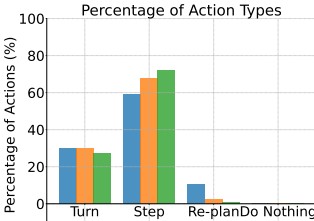

(a) Comparison of model action types in static obstacle environments (success cases analysis).

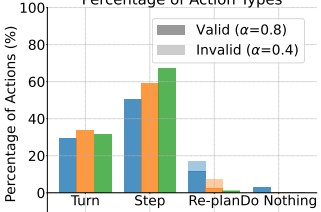

(b) Comparison of model action types in dynamic obstacle (pedestrian) environments (success cases analysis).

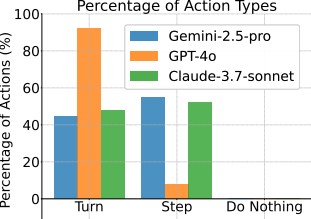

(c) Comparison of model action types without local action planner (all cases analysis).

Figure 15: **Distribution of Action Types for Navigation Models in Environments with/without Obstacles and Different Obstacle Types.**

*Model capability is a decisive factor in task success.* State-of-the-art models such as `GPT-4o`, `Claude-3.7-Sonnet`, and `Gemini-2.5-Pro` consistently outperform the rule-based baseline across all three difficulty levels. These advanced MLLMs achieve higher SR, along with lower CC and CC-S, indicating their ability to understand the environment and avoid obstacles more effectively. In contrast, `GPT-4o-mini` performs even worse than the rule-based baseline, with lower SR and higher CC and CC-S, suggesting that it struggles to follow task rules and may be prone to hallucinations or unreasonable actions.

*MLLMs demonstrate limited spatial reasoning.* Although larger MLLMs attain higher SR across all tasks, we do not observe a corresponding drop in CC or CC-S. This implies that while these models attempt to avoid obstacles, they often fail—frequently grazing or barely touching the edges of objects. This behavior suggests that current MLLMs possess strong visual recognition abilities but lack a robust understanding of spatial relationships and depth.

*Different models exhibit distinct strengths.* Our analysis reveals that individual models have specialized advantages. For instance, `GPT-4o` and `Claude-3.7-Sonnet` show higher STR, indicating that its failures are more often due to getting stuck rather than poor decisions, suggesting better decision-making but weaker spatial adaptability. In contrast, `Gemini-2.5-Pro` shows a lower STR, implying that it struggles more with planning and timely decision-making. These trends are consistent with their respective SR, CC, and CC-S values: `GPT-4o` and `Claude-3.7-Sonnet` excel in making quick and precise decisions, while `Gemini-2.5-Pro` shows better spatial comprehension. Figure 15a shows that `Gemini-2.5-Pro` performs "choose a new waypoint" more than other models, which further explains that `Gemini-2.5-Pro` are more sensitive to obstacle avoidance. Additionally, lower NDC supports these findings by showing which models complete tasks using fewer steps, reflecting planning efficiency.

*MLLMs fail to function as truly embodied agents.* Although MLLM-driven agents do not achieve 100% RVR in the table, additional experiments on the Hard tasks—where all pedestrian signals were kept red—reveal a more fundamental issue: agents crossed streets despite the red lights. To correctly perceive the signal state, an agent must look up and focus on the signal. Upon detecting a red light, it should repeatedly choose the *do nothing* action until the signal turns green. However, experimental results show that none of the agents chose to adjust their view, and very few opted to do nothing, indicating that MLLMs do not recognize their ability to adjust visual sensors. Instead, they react only based on the input they are given. This suggests that LLM-based agents require substantial additional components to behave like real embodied agents—the LLM alone is not sufficient.

**Dynamic Obstacles.** Table 5 presents the performance of different models on dynamic navigation tasks. `GPT-4o` and `Claude-3.7-Sonnet` demonstrate superior performance, achieving higher SR and lower NDC. This suggests that these models are capable of making faster and more decisive actions, enabling them to complete navigation tasks efficiently in dynamic environments. In contrast, `Gemini-2.5-Pro` experiences a significant drop in SR and exhibits higher NDC values—comparable to those of `GPT-4o-mini`—while also recording the lowest values in both CC and CC-S among all baselines.

Figure 15b reveals that `Gemini-2.5-Pro` frequently switches waypoints and tends to choose valid ones, suggesting stronger spatial sensitivity. However, this caution often leads to hesitation or

stalls near moving obstacles, highlighting a trade-off between spatial adaptability and robust task completion.

Table 5: **Performance Comparison under Dynamic Obstacles.** SR: Success Rate, CC: Collision Count, CC-S: Collision Count (Success Only), NDC: Normalized Decision Count. Human: Human Collision Count, Obj: Object Collision Count.

| Baseline | SR↑ | CC↓ | | | CC-S↓ | | | NDC↓ |
|---|---|---|---|---|---|---|---|---|
| | | Human | Obj | All | Human | Obj | All | |
| Rule-based | 80.00 | 2.93 | 2.97 | 5.90 | 2.54 | 2.83 | 5.38 | N/A |
| GPT-4o-mini | 26.67 | 1.37 | 3.80 | 5.17 | 1.12 | 2.75 | 3.88 | 2.43 |
| GPT-4o | 86.67 | 1.13 | 2.63 | 3.77 | 1.04 | 2.54 | 3.58 | 2.30 |
| Claude | **93.33** | 1.00 | 2.57 | 3.57 | 1.04 | 2.36 | 3.39 | **2.27** |
| Gemini | 56.67 | **0.33** | **1.60** | **1.93** | **0.35** | **1.18** | **1.53** | 2.77 |

Table 6: **Performance Without Local Action Planner.** SR: Success Rate, CC: Collision Count, CC-S: Collision Count (Success Only), DSS: Decision Steps in Success. Bldg: Building Collision Count, Obj: Object Collision Count.

| Baseline | SR↑ | CC↓ | | | CC-S↓ | | | DSS↓ |
|---|---|---|---|---|---|---|---|---|
| | | Bldg | Obj | All | Bldg | Obj | All | |
| GPT-4o | 0.00 | 4.50 | 0.00 | 4.50 | N/A | N/A | N/A | N/A |
| Claude | 56.67 | 18.50 | 1.63 | 20.13 | 11.71 | 2.18 | 13.88 | **58.71** |
| Gemini | **60.00** | **15.83** | **0.40** | **16.23** | **10.44** | **0.50** | **10.94** | 68.72 |

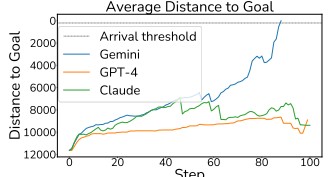

Figure 16: **Step-wise Average Destination Distance across Tasks.** `GPT-4o` often exhibits early stagnation, making limited goal progress, while `Claude-3.7-Sonnet` shows fluctuating effectiveness. `Gemini-2.5-Pro` stands out by demonstrating the most stable and continuous reduction in goal distance, highlighting its superior planning and navigation capabilities without a local action planner.

## B.8  Ablation Study: Without Local Action Planner

To evaluate the robustness of MLLM-driven agents under more challenging conditions, we remove the local action planner and re-evaluate their performance on Medium-level tasks. In this setting, agents are only provided with the final destination instead of a pre-defined series of waypoints. They are required to not only avoid collisions but also plan an efficient trajectory autonomously. All agents are restricted to a maximum of 90 decision steps per task.

We compare three state-of-the-art MLLMs: `GPT-4o`, `Claude-3.7-Sonnet`, and `Gemini-2.5-Pro` (gemini-2.5-pro-preview-05-06). As shown in Table 6, `GPT-4o` fails to complete any task successfully (SR = 0), while `Claude-3.7-Sonnet` achieves a 56.67% SR and it also achieves the lowest DSS of 58.71, indicating more efficient decision-making. `Gemini-2.5-Pro` outperforms both with the highest SR (60.0%) and the lowest overall CC (16.23), including significantly fewer building and object collisions.

Figure 16 illustrates the distance-to-goal progression for each model from a single experimental run. The behavioral patterns underlying these distance trends are further elucidated by Figure 15c, which shows `GPT-4o` and `Claude-3.7-Sonnet` frequently engaging in non-progressive turning actions, whereas `Gemini-2.5-Pro` predominantly maintain forward movement.

## B.9  Chain of Thought Example

This section, we showcase a representative chain of thought from `Gemini-2.5-Pro` as it navigated the Easy level task. This demonstration is provided to give readers a transparent view of the model's deliberative process when prompted to "think step-by-step." The following transcript captures the VLM's sequential reasoning, from understanding the prompt to formulating its next action.

---
**Example Chain of Thought from `Gemini-2.5-Pro` on a Successful Easy Task**

```
Prompt: You are currently at ...
Action: Turn around
Reasoning: The relative angle to the next waypoint is 144.77 degrees, which is
    greater than 15 degrees. So I need to turn to face the next waypoint. The
    path to the waypoint is clear based on the images.
```
---

```
Prompt: You are currently at ...
Action: Step forward
Reasoning: I am facing the next waypoint (relative angle 0.77 degrees) and the
    path is clear. My last action was a turn, so now I should move forward. The
     distance to the waypoint is 948.68 cm, and my speed is 200 cm/s, so I can
    reach it in 4.74 seconds, which is within the allowed duration.

Prompt: You are currently at ...
Action: Turn around
Reasoning: The relative angle to the next waypoint is -16.89 degrees. Since the
    absolute value of this angle (16.89) is greater than the 15-degree
    threshold, I need to adjust my direction. The negative sign indicates the
    waypoint is to my left, so I will turn counter-clockwise. The current view
    shows no obstacles directly in the path to the waypoint. My last action was
     stepping forward, so a turn action is not repetitive.

Prompt: You are currently at ...
Action: Step forward
Reasoning: I am facing the next waypoint (relative angle -0.89 degrees). The
    path is clear. My last action was turning, so I should move forward now.
    The distance to the waypoint is 896.46 cm, and my speed is 200 cm/s, so the
     time to reach is 896.46 / 200 = 4.48 seconds.

Prompt: You are currently at ...
Action: Step forward
Reasoning: The path to the next waypoint is clear and I am facing it. The
    relative angle is small (-2.55 degrees), so no turn is needed. I will step
    forward to get closer to the waypoint.

Prompt: You are currently at ...
Action: Step forward
Reasoning: The path to the next waypoint is clear, and the relative angle is
    small (-7.56 degrees). Therefore, I will step forward towards the waypoint.
     The time to reach the waypoint at 200 cm/s is approximately 4.5 seconds,
    which is within the allowed duration.

Prompt: You are currently at ...
Action: Turn around
Reasoning: My current target waypoint is Vector(x=-1850.0, y=-4800.0), which I
    selected in my previous action. The relative angle to this waypoint is
    179.06 degrees. This is significantly greater than the 15-degree threshold,
     indicating I am facing almost directly opposite to the waypoint. The
    current visual information shows a tree immediately in front of me,
    blocking my current path. By turning 179.06 degrees to the right (clockwise
    , as the relative angle is positive), I will align myself with the waypoint
    . The tree currently in front will then be behind me. According to the
    guidelines, if the last action was choosing a new waypoint, I should
    attempt to move towards it. Turning to face it is the first step. This
    action does not violate rules against repeated waypoint selection or
    consecutive turns, as my action history is CHOOSE_WAYPOINT, TURN,
    CHOOSE_WAYPOINT, and now TURN.

Prompt: You are currently at ...
Action: Step forward
Reasoning: I am currently facing the next waypoint (Vector(x=-1850.0, y=-4800.0)
    ) with a relative angle of 0.06 degrees, which is well within the 15-degree
     tolerance. The current view shows that the path to the waypoint is clear.
    My last action was a turn, and prior to that I chose a new waypoint and
    turned. To make progress and avoid repeated turns, I should now move
    forward. The next waypoint is 118.72 cm away. The instructions state that a
     step duration of 1-5 seconds is acceptable. Therefore, I will choose to
    step forward for 1.0 second. At a speed of 200 cm/s, this means I will move
     200 cm, which will overshoot the current waypoint. This is acceptable
    given the duration constraint, and the system will provide a new state
    afterwards.

Prompt: You are currently at ...
Action: Turn around
Reasoning: The relative angle to the next waypoint is 77.64 degrees, which is
    greater than 15 degrees. Therefore, I need to turn to face the waypoint.
    Since the angle is positive, I will turn right (clockwise).

Prompt: You are currently at ...
Action: Choose waypoint
Reasoning: The current next waypoint Vector(x=-1550.0, y=-3900.0) is blocked by
    a barrier visible in the current view. I am choosing a new waypoint Vector(
    x=-1250.0, y=-3900.0), which is to the right of the current blocked path.
    Visual inspection suggests this path might be clearer.
```

Prompt: You are currently at ...
Action: Turn around
Reasoning: My last action was choosing a new waypoint. The current waypoint is
    Vector(x=-1250.0, y=-3900.0). The relative angle to this waypoint is -15.62
    degrees. Since the absolute value of the relative angle (15.62 degrees) is
    greater than the 15-degree threshold, I need to turn to face the waypoint.
    The negative angle indicates the waypoint is to my left, so I will turn
    counter-clockwise by 15.62 degrees. The current view shows no immediate
    obstacles in that direction.

Prompt: You are currently at ...
Action: Turn around
Reasoning: The last action was choosing a new waypoint. The current waypoint is
    Vector(x=-1550.0, y=-3900.0). The relative angle to this waypoint is 15.64
    degrees, which is greater than the 15-degree threshold for turning. The
    waypoint is to my right, so I need to turn clockwise. The path to the
    waypoint appears clear from the current view.

Prompt: You are currently at ...
Action: Step forward
Reasoning: I am facing the next waypoint (relative angle 0.64 degrees) and it is
    not blocked. My last action was a turn, and before that I chose a new
    waypoint. Therefore, I should now move towards the current waypoint. The
    distance is 928.38 cm and speed is 200 cm/s, so it would take about 4.64
    seconds to reach it. I will step forward for 4.5 seconds.

Prompt: You are currently at ...
Action: Choose waypoint
Reasoning: The current next waypoint Vector(x=-1550.0, y=-3900.0) is visibly
    blocked by a construction barrier. I am choosing a new waypoint, Vector(x
    =-1850.0, y=-3900.0), which is to the left of the current blocked waypoint
    and appears to offer a clear path around the barrier based on the current
    view.

Prompt: You are currently at ...
Action: Choose waypoint
Reasoning: My last action was choosing a new waypoint. The current next waypoint
    Vector(x=-1850.0, y=-3900.0) is 19.41 degrees to my right. The current
    view shows a barrier to the right, which likely blocks this waypoint.
    Therefore, I will choose a new waypoint that is to the left of the current
    one and the barrier, which is Vector(x=-2300.0, y=-3900.0).

Prompt: You are currently at ...
Action: Step forward
Reasoning: I am facing the next waypoint (relative angle 0.78 degrees), and the
    path appears clear based on the current and previous views. My last action
    was a turn, preceded by choosing a new waypoint. According to the rules, I
    should now move towards the current waypoint. I will step forward for 5.0
    seconds to get closer to the waypoint, which is 1046.26 cm away and my
    speed is 200 cm/s.

Prompt: You are currently at ...
Action: Turn around
Reasoning: My last action was choosing a new waypoint. The current waypoint is
    not blocked. The relative angle to the waypoint is -24.59 degrees, which is
    greater than 15 degrees. Therefore, I need to turn left to face the
    waypoint.

Prompt: You are currently at ...
Action: Choose waypoint
Reasoning: The current waypoint Vector(x=-1850.0, y=-3900.0) is blocked by a
    construction barrier visible in the current view. Therefore, a new waypoint
    needs to be chosen. Vector(x=-1550.0, y=-3900.0) is selected as it is to
    the right of the barrier, requires a moderate turn (+19.3 degrees), and
    progresses towards the final destination. The right side appears more open
    based on the image.

Prompt: You are currently at ...
Action: Turn around
Reasoning: My last action was choosing a new waypoint. The relative angle to the
    current waypoint is -19.36 degrees, which is greater than 15 degrees.
    Therefore, I need to turn to face the waypoint. The obstacle in the view is
    distant and does not immediately block the path for turning.

Prompt: You are currently at ...
Action: Turn around
Reasoning: The last action was choosing a new waypoint. The current waypoint is
    not blocked based on the visual information. The relative angle to the next

```
            waypoint is -16.62 degrees, which is greater than the 15-degree threshold
            for turning. Therefore, I will turn left by 16.62 degrees to align with the
            next waypoint.

Prompt: You are currently at ...
Action: Step forward
Reasoning: The next waypoint is 441.28 cm away and the relative angle is -1.47
            degrees, which is small. The path appears clear from the current and
            previous views. The last action was stepping forward, so continuing to move
            towards the waypoint is appropriate. The calculated duration to reach the
            waypoint is 2.2 seconds, which is within the acceptable range.

Prompt: You are currently at ...
Action: Turn around
Reasoning: The relative angle to the next waypoint is 47.34 degrees, which is
            greater than 15 degrees. Therefore, I need to turn to face the waypoint.
            The waypoint is to my right, so I will turn clockwise.

Prompt: You are currently at ...
Action: Step forward
Reasoning: My last action was to turn towards the waypoint. I am now facing the
            waypoint (relative angle 0.34 degrees). The path to the waypoint appears
            clear in both current and previous views. The distance to the waypoint is
            939.89 cm. At a speed of 200 cm/s, it would take about 4.7 seconds. I will
            step forward for 4.5 seconds, which is within the 1-5 second limit and will
            get me close to the waypoint.

Prompt: You are currently at ...
Action: Step forward
Reasoning: The relative angle to the next waypoint is -10.52 degrees, which is
            within the 15-degree tolerance, so I am roughly facing it. The current view
            shows a clear path to the waypoint. My last action was stepping forward,
            so I should continue moving towards the waypoint. The distance to the
            waypoint is 1347.75 cm. Stepping forward for 4.0 seconds at 200 cm/s will
            cover 800 cm, bringing me closer to the waypoint without exceeding the
            maximum step duration of 5 seconds.
```

Figure 17: **Example Chain of Thought from** `Gemini-2.5-Pro` **on a Successful Easy Level Task.**

## C Case Study 2 Details

### C.1 Problem Formulation

In Case Study 2, we formalize the scenario as a sequential decision-making problem within a multi-agent environment. Specifically, we focus on evaluating the decision-making capabilities of *Delivery Agents*. *Customers* and *Stores* are managed by a unified rule-based manager, while each *Delivery Agents* autonomously decides actions to maximize profit.

#### C.1.1 Agent Definition

Let the set of *Delivery Agent* be represented as $\mathcal{D} = \{D\_1, D\_2, \ldots, D\_n\}$, each operating independently yet concurrently in the shared environment.

#### C.1.2 Decision Variables

At each discrete timestep $t$, an agent $D\_i$ chooses from the following set of actions:

1. Accept or reject available orders.
2. Plan routing along waypoints defined by a provided map structure.
3. Purchase transportation equipment to enhance delivery efficiency, subject to available financial resources.
4. Adjust travel speed $v\_t \in [v\_min, v\_max]$ to balance energy consumption and delivery speed.

#### C.1.3 Constraints

Each *Delivery Agent* $D\_i$ operates under the following constraints:

- **Energy Constraint**: Agent $D\_i$ has an energy level $E\_t \geq 0$. Movement at higher speeds consumes more energy, and energy can be replenished by purchasing beverages at supply points.
- **Financial Constraint**: Each agent starts with an initial amount of money $M\_0$. Actions such as purchasing transportation equipment or beverages require spending money. The agent's financial status at time $t$ is represented by $M\_t \geq 0$.
- **Spatial Constraint**: Movements must follow a predetermined waypoint map $\mathcal{G} = (\mathcal{V}, \mathcal{E})$, where $\mathcal{V}$ is the set of waypoints, and $\mathcal{E}$ represents feasible paths.

#### C.1.4 Objective Function

The primary objective for each *Delivery Agent* $D\_i$ is to maximize total money $M\_T$ accumulated by the end of simulation period $T$:

$$\max \quad M\_T = M\_0 + \sum\_t = 1^T (R\_t - C\_t) \tag{1}$$

where:

- $R\_t$ represents revenue earned at timestep $t$ from successful deliveries.
- $C\_t$ represents costs incurred at timestep $t$, including equipment purchase costs, energy replenishment costs, and penalties from delayed or failed deliveries.

#### C.1.5 Agent Dynamics

Agent dynamics at timestep $t$ are governed by:

$$E\_t + 1 = E\_t - f(v\_t) + \Delta E\_replenish, \tag{2}$$
$$M\_t + 1 = M\_t + R\_t - C\_t, \tag{3}$$
$$X\_t + 1 = \text{WaypointUpdate}(X\_t, a\_t), \tag{4}$$

where:

- $f(v\_t)$ denotes energy consumption as a function of travel speed.

- $\Delta E$\_replenish is the energy replenished through beverages at supply points.
- $X$\_$t$ denotes the agent's position at timestep $t$, updated based on actions $a$\_$t$ following the waypoint system.

This formulation offers a comprehensive framework for evaluating long-horizon decision-making strategies of *Delivery Agents* within the multi-agent environment of SIMWORLD.

### C.1.6 Metrics Formulation

Aligned with our hierarchical evaluation framework, we define the following metrics for each agent $D_i$ operating over $T$ timesteps:

**High-level: Total Profit**

$$P_i \;=\; M_i(T) \;-\; M_i(0)\,,$$

where $M_i(t)$ is the agent's monetary balance at time $t$.

**Mid-level: Operational Effectiveness**

$$\text{Order Success Rate:} \quad \text{SR}_i = \frac{|\mathcal{O}_i^{\text{succ}}|}{|\mathcal{O}_i^{\text{bid}}|}\,, \tag{5}$$

$$\text{Delay Rate:} \quad \text{DR}_i = \frac{|\mathcal{O}_i^{\text{delay}}|}{|\mathcal{O}_i^{\text{succ}}|}\,, \tag{6}$$

$$\text{Energy Efficiency:} \quad \text{EE}_i = \frac{P_i}{\sum_{t=1}^{T} e_i(t)}\,, \tag{7}$$

$$\text{Order-Sharing Number:} \quad \text{SH}_i = |\mathcal{O}_i^{\text{shared\_succ}}|,, \tag{8}$$

$$\text{Investment Number:} \quad \text{IN}_i = |\mathcal{A}_i^{\text{buy\_bike\_succ}}|,, \tag{9}$$

where

- $\mathcal{O}_i^{\text{bid}}$ is the set of orders on which $D_i$ bid,
- $\mathcal{O}_i^{\text{succ}} \subseteq \mathcal{O}_i^{\text{bid}}$ are those bids that led to successful deliveries,
- $e_i(t)$ is energy consumed by $D_i$ at timestep $t$,
- $\mathcal{O}_i^{\text{delay}} \subseteq \mathcal{O}_i^{\text{succ}}$ are deliveries completed after the deadline,
- $\mathcal{O}_i^{\text{shared\_succ}}$ denotes successfully shared orders.
- $\mathcal{A}_i^{\text{buy\_bike\_succ}}$ demotes successfully buy bike actions.

**Low-level: Instruction and Rule Compliance**

$$\text{IDR}_i = \frac{I_i}{D_i}\,, \quad \text{IDR}_{i,a} = \frac{I_{i,a}}{D_{i,a}} \quad \forall a \in \{\text{bid, pickup, deliver, beverage, shared, meet, cancel, speed, nothing}\},$$

where

- $D_i$ is the total number of actions taken by $D_i$,
- $I_i$ is the total number of those actions deemed invalid,
- $D_{i,a}$ and $I_{i,a}$ are the counts of attempts and invalid attempts for action $a$.

For each metric $m \in \{P, \text{SR}, \text{ADS}, \text{DR}, \text{EE}, \text{SH}, \text{IDR}, \text{IDR}_a\}$, we report group-level aggregates:

$$\overline{m} \;=\; \frac{1}{n}\sum_{i=1}^{n} m_i, \quad \text{median}(m), \quad \sigma(m) \;=\; \sqrt{\frac{1}{n-1}\sum_{i=1}^{n}(m_i - \overline{m})^2}\,.$$

These statistics are written to our CSV outputs for each model grouping. And we show the part of metric in the Table 3

## C.2 Action Space

The low-level action space comprises fine-grained executable commands in Unreal Engine, such as move forward, rotate, pick up, and drop off. In contrast, high-level actions are determined by the LLM.

Table 7: **Hierarchical Action Space Design in Case Study 2.** High-level actions are given to language models to make decision, which correspond to strategic decisions, while low-level actions are only exposed to local action planner module to execute concrete movements and interactions.

| Action Level | Action Name | Description | Invocation Method |
|---|---|---|---|
| High-Level Actions | Bid Order | Offer a price to a new order on platform to compete with other Model Generations | Model Generation |
| | Pick Up Order | Navigate to the pick-up point of order | Model Generation |
| | Deliver Order | Navigate to the delivery point and complete the order | Model Generation |
| | Share Order | Publish the order for multi-model generation cooperation | Model Generation |
| | Cancel Share Order | Cancel a shared order that has been published | Model Generation |
| | Go to Meet-point | Navigate to the meet point for the shared order | Model Generation |
| | Purchase Scooter | Buy and use a scooter | Model Generation |
| | Purchase Drinks | Buy consumables to restore energy | Model Generation |
| | Adjust Speed | Adjust travel speed | Model Generation |
| Low-Level Actions | Move Forward | Basic movement action | Local Planner |
| | Stop | Stop moving | Local Planner |
| | Rotate | Adjust the facing direction | Local Planner |
| | Change Speed | Adjust walking speed | Local Planner |
| | Drive Scooter | Control a scooter for movement | Local Planner |

## C.3 Prompt Design

---

**System Prompt of Long-Horizon Delivery Task**

```
SYSTEM_PROMPT = """
You face the following considerations:
1. Accepting orders along your route can increase earnings efficiency by
     reducing empty travel time and grouping deliveries.
2. Risks include unpredictable road conditions, inaccurate time estimates, and
     potential delays leading to penalties and lower ratings.
3. Late deliveries incur negative ratings and monetary penalties, affecting your
     overall performance.
4. For each order, evaluate its fit to your current route, delivery time promise
     , and balance efficiency with service quality.
5. You can buy a bike for <PRICE_OF_BIKE> dollars to increase speed and reduce
     energy consumption.
6. You can change your speed (<DELIVERY_MAN_MIN_SPEED> to <
     DELIVERY_MAN_MAX_SPEED> cm/s); higher speeds consume more energy.
7. You may share a picked-up order with another delivery man by specifying a
     meeting point; both delivery men share earnings and penalties if late. This
     can bring both of you more money and save your energy.
8. You can cancel a shared order if no one accepts it after a long wait. That
     means you decide to do it by yourself, rather than waiting for too long
     which may lead to a penalty.
9. If low on energy, go to a supply point to buy a beverage (cost <
     COST_OF_BEVERAGE> dollars, recovers <DELIVERY_MAN_RECOVER_ENERGY_AMOUNT>
     energy).
10. You need to finish order in time to make money, do not bid a order and not
     delivery it.

Important:
- All your decisions are based on your persona and the considerations.
- Provide only the final action in JSON, the JSON format would be given in each
     round of request. Do not add explanations or repeat given examples.
- There are other delivery men in the city, try to compete with them or
     cooperate with them according to the your persona and the situation.
- If you can not make the most money at the end of game, you will die. So try
     your best to use the rules to make the most money.
"""
```

Figure 18: **Examples System Prompt of Long-horizon Delivery Task.**

```
# User prompt to output the reasoning
REASONING_USER_PROMPT = """
You are now at <POSITION> in a city, where the unit is cm.
You can buy and use a beverage to recover your energy.
Your position is always at or near a node in the graph and you can move from one
    node to another node only if there is an edge between the two nodes. All
    your possible next waypoints are:
<POSSIBLE_NEXT_WAYPOINTS>
You have already picked up the following order in the format of Order:
<PICKED_UP_ORDER>
You already have the following orders in the format of List[Order]:
<ORDERS>
Also, you can bid for new orders from the platform in the format of List[Order]:
<ORDERS_TO_BID>
And you could notice the following notification, which is the platform wanna you
    to know:
<NOTIFICATION>
For each order, you should check the attributes before making a decision:
- max_sale_price: the maximum price of the order.
- min_sale_price: the minimum price of the order.
- customer_position: the position of the customer.
- store_position: the position of the store.
- has_picked_up: whether the order has been picked up by you.
- estimated_time: the estimated time to deliver the order.
- is_shared: whether the order is shared.
- meeting_point: the meeting point of the shared order.
- spent_time: the time that has passed since the order was opened.
Note that you consume energy as long as you move. There are two ways to move:
    walking and driving. You lose energy as long as you move. Walking is your
    default way to move and you can choose to buy a bike to drive if you have
    enough money. Your current walking speed is <SPEED> cm/s. Faster walking
    speed will consume more energy. Now your moving way is <PHYSICAL_STATE>.
Currently, you have <MONEY> dollars and <ENERGY> energy.

You have the following history of your actions:
<HISTORY>
The first item of each tuple in the history is your position, and the second
    item is your action.
From top to bottom, the history is sorted by time. The most recent action is at
    the bottom.
To improve efficiency, avoid visiting the same position more than once unless
    necessary and avoid repeating the same action except Deliver a order.

Now you need to make a decision. You have the following options:

1. Bid for an order. You can bid for an order from the platform. You can only
    bid for an order on the platform. The bid price should be between the
    min_sale_price and max_sale_price of the order.
2. Pick up an order. You can pick up an order from your orders. You can only
    pick up an order that has not belonged to you.
3. Deliver an order. You can deliver an order from your orders. Do not deliver
    the order that has not been picked up by you.
4. Buy a beverage. When your energy is low, you can buy a beverage to recover
    your energy. The price of beverage start from <COST_OF_BEVERAGE> dollars
    and will go up 5% every time you buy a beverage. A beverage recovers <
    RECOVER_ENERGY_AMOUNT> energy.
5. Open a shared order. You can open a shared order from your orders so another
    delivery man can join the shared order. This can save your time and energy,
     but you need to split the earnings with another delivery man. You can only
     share an order that has been picked up by you. You need to decide the
    meeting point and go there to wait for another delivery man. They will
    finish the rest of the order. Note that if the shared order is delivered
    late, both of you will receive negative customer ratings and penalties.
6. Go to the meeting point. If you are joining a shared order, you need to go to
     the meeting point and wait for another delivery man.
7. Cancel a shared order. If a shared order has not been accepted by another
    delivery man for a long time, you can cancel it and do it by yourself.
8. Change walking speed. You can change your walking speed to a new walking
    speed (range from 100 to 250 cm/s).
9. Buy a bike. You can buy a bike to increase your speed. A bike costs <
    PRICE_OF_BIKE> dollars. But it will increase your speed and decrease your
    energy consumption.

Make a reasoning about your next action based on the above information. For each
    action, you have the following tips:
```

```
      Explain your reasoning using one sentence with clear instructions in json format
          .

      Example reasoning (do not repeat):

      {'reasoning': 'I have enough money to buy a bike, so I will buy a bike now.'}

      Now, provide only the action JSON with keys: reasoning.
      """

      # User prompt to output the action
      ACTION_USER_PROMPT = """
      Your reasoning: <REASONING>

      Now you need to respond your action in json format. Note that you can only
          choose one of the following actions:

      1. Bid for an order. Respond with the index of the order you want to bid for and
           the bid price.
      2. Pick up an order. Respond with the index of the order you want to pick up,
          the next waypoint and the target point.
      3. Deliver an order. Respond with the index of the order you want to deliver,
          the next waypoint and the target point.
      4. Buy a beverage. Just respond with your choice.
      5. Open a shared order. Respond with the index of the order you want to share
          and the meeting point.
      6. Go to the meeting point. Respond with the index of the shared order you want
          to join and the next waypoint.
      7. Cancel a shared order. Respond with the index of the shared order you want to
           cancel.
      8. Change walking speed. Respond with the new walking speed.
      9. Buy a bike. Just respond with your choice.

      Provide only the action JSON with keys: choice, index, target_point,
          meeting_point, next_waypoint, new_speed.

      Example action (do not repeat):

      {'choice': 1, 'index': 1, 'target_point': (200, 200), 'meeting_point': (300,
          300), 'next_waypoint': (400, 400), 'new_speed': 200}
      """
```

Figure 19: **Examples User Prompt of Long-horizon Delivery Task.**

## C.4   Dataset Details

### C.4.1   Simulated Environment Generation

Maps are created via a multi-stage procedural pipeline (used in both variants):

1. **Road-Grid Skeleton:** define an $N \times N$ road network ($N \in \{3, \ldots, 8\}$), centered at $(0, 0)$, with uniform segment length and spacing.
2. **Element Population:** place buildings programmatically along each road; add trees and street lights for visual realism.
3. **Data Screening:** export all scene elements, then remove non-restaurant nodes (e.g., hospitals, schools) to prepare for real-world alignment.

### C.4.2   Real-World Dataset (LLMDeliveryReal-100)

This variant aligns simulated blocks with actual restaurant data:

1. **Data Acquisition:** fetch Manhattan restaurant listings (latitudes 40.758–40.770, longitudes –73.995– –73.970) via Google Maps API; clean and convert to geospatial points.
2. **Neighborhood Assignment:** build polygonal blocks from street intersections; assign each restaurant via point–in–polygon tests.

3. **Normalization:** map real streets/avenues to grid indices $0, \ldots, N-1$. For each restaurant, compute relative position $(\alpha, \beta) \in [0,1]^2$ within its block and obtain

$$(x_{\mathrm{norm}}, y_{\mathrm{norm}}) = (x_0 + \alpha,\ y_0 + \beta).$$

4. **Building Coordinate Scaling:** convert original building coordinates $(x_{\mathrm{orig}}, y_{\mathrm{orig}}) \in [-\textit{offset}, \textit{offset}]^2$ to $[0, N-1]^2$ via

$$x_{\mathrm{norm}} = \frac{x_{\mathrm{orig}} + \textit{offset}}{2\textit{offset}}\,(N-1), \quad y_{\mathrm{norm}} = \frac{y_{\mathrm{orig}} + \textit{offset}}{2\textit{offset}}\,(N-1).$$

5. **Matching:** compute pairwise Euclidean distances in normalized space; match each building to its nearest restaurant within threshold 0.25, resolving conflicts by closest-pair selection.

6. **Metadata Injection:** embed a match flag and restaurant metadata (ID, name, category, geo-coords) into each building node.

### C.4.3  Procedurally-Generated Dataset (LLMDeliveryRandom-1K)

This variant omits real-world alignment and instead samples agent and store positions randomly:

- **Generation Type:** "random procedural generation."
- **Agent Placement:** DeliveryManager instantiates fixed counts of stores, customers and delivery agents at uniformly random building locations.

### C.4.4  File Structure

Each map folder contains:

- `metadata.json`: difficulty, entity counts, generation type, file paths.
- `positions.json`: 2D coordinates of stores, customers, delivery agents.
- `progen_world.json`: full scene graph with normalized coords and match flags.
- `roads.json`: road segment list with endpoints.
- `city_visualization.png`: full-view rendering.
- `dm_visualization.png`: delivery-focused schematic.

### C.5  Results

### C.5.1  Main Results

`DeepSeek-V3` $(69.475 \pm 16.772)$ and `Claude-3.5-Sonnet` $(69.068 \pm 20.685)$ achieved the highest mean profits, with `Claude-3.5-Sonnet` also leading in mean successful orders $(2.733 \pm 1.102)$ and energy efficiency $(0.5411 \pm 0.1981)$. Notably, these superior average outcomes were associated with substantial performance variability, as reflected in their respective standard deviations.

Conversely, `Gemini-2.5-Flash`, while attaining a moderate mean profit of $42.423$, exhibited markedly more consistent profit generation with a standard deviation of $\pm 3.103$, and also demonstrated stability in successful orders $(2.100 \pm 0.173)$. Extreme variability was evident in specific metrics for certain models; for instance, sharing counts for `Deepseek-Prover-V2` $(7.333 \pm 8.386)$ and `Claude-3.5-Sonnet` $(11.333 \pm 8.386)$ showed standard deviations exceeding their means, indicating highly unpredictable behavior in this aspect.

The `GPT-4o-mini` model consistently yielded zero values across all metrics $(0.000 \pm 0.000)$, suggesting it does not truly understand the goals to make reasonable decisions based on the given instructions and context information.

Higher investment strategies, such as those adopted by `DeepSeek-V3` $(8.000 \pm 3.000)$ and `Claude-3.5-Sonnet` $(9.000 \pm 3.464)$, generally correlated with greater mean profit achievement but also with increased outcome volatility. These findings underscore a prevalent trade-off between optimizing for peak average performance metrics and ensuring consistent, predictable agent behavior, a critical consideration for robust deployment in dynamic environments.

### C.5.2  Environment Configuration

We further investigate how different environmental configurations impact agent behavior and overall performance. Specifically, we explore two key factors: the global order availability and the agents'

initial financial endowment. For each factor, we conduct a series of controlled experiments to observe how variations affect agents' action distributions.

As shown in Figure 20a, when the total number of available orders increases, agents tend to perform fewer pickup and delivery actions and instead choose the `do nothing` action more frequently. This suggests that in resource-rich environments, agents are more inclined to conserve energy and avoid unnecessary effort, opting to wait for optimal opportunities rather than actively pursue deliveries. Conversely, in low-resource settings, agents are more motivated to engage in delivery tasks to secure profits. Additionally, as resource abundance increases, agents demonstrate a higher tendency to initiate and complete shared deliveries, likely as a means to reduce energy costs through collaboration.

Figure 20b illustrates the impact of agents' initial monetary resources. As initial capital increases, agents are less reliant on aggressive bidding and instead prioritize actions such as order pickup. When funds are limited, competition intensifies, leading to more frequent bidding behavior. Furthermore, with sufficient initial capital, agents are more willing to invest in infrastructure—such as purchasing a bike—which enhances their long-term delivery efficiency.

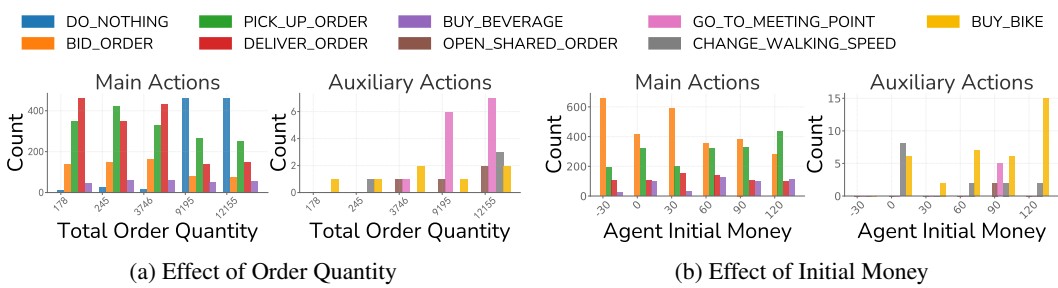

(a) Effect of Order Quantity           (b) Effect of Initial Money

Figure 20: **Action Distribution across Environmental Settings.** (a) Shows the change of effect of global order quantity on agent behavior. (b) Shows the effect of initial money on action selection.

### C.5.3 Influence of Persona.

Personality traits significantly affect the decision-making and performance of delivery agents. As shown in Figure 21, agents with higher Conscientiousness tend to exhibit a lower frequency of bidding actions, a higher frequency of task-completion actions (e.g., picking up orders), and achieve a higher bid win rate. This suggests that conscientious agents prioritize task completion over strategic competition. Agents with higher Agreeableness are less likely to remain inactive (i.e., perform "do nothing" actions) and tend to achieve higher bid win rates. Conversely, agents with lower agreeableness display higher inactivity and narrower bidding price ranges, limiting their competitiveness. Interestingly, agents with higher Openness exhibit reduced engagement in delivery tasks, possibly because they explore competitive or unconventional bidding strategies that divert attention from task execution.

### C.6 Chain of Thought Example

This section, we showcase a representative chain of thought from `Claude-3.5-Sonnet`. Which shows the model's reasoning process under different circumstances.

---

**Example Chain of Thought from `Claude-3.5-Sonnet` on different actions**

```
Action: Do Nothing
Reasoning:
Case1= """
Since I have no orders to deliver and no new orders to bid on, I'll wait at my
    current position to conserve energy and wait for new orders to appear.
"""

Case2= """
I have no orders to deliver and no money to buy anything, so I should wait for
    new orders from the platform.
"""
```

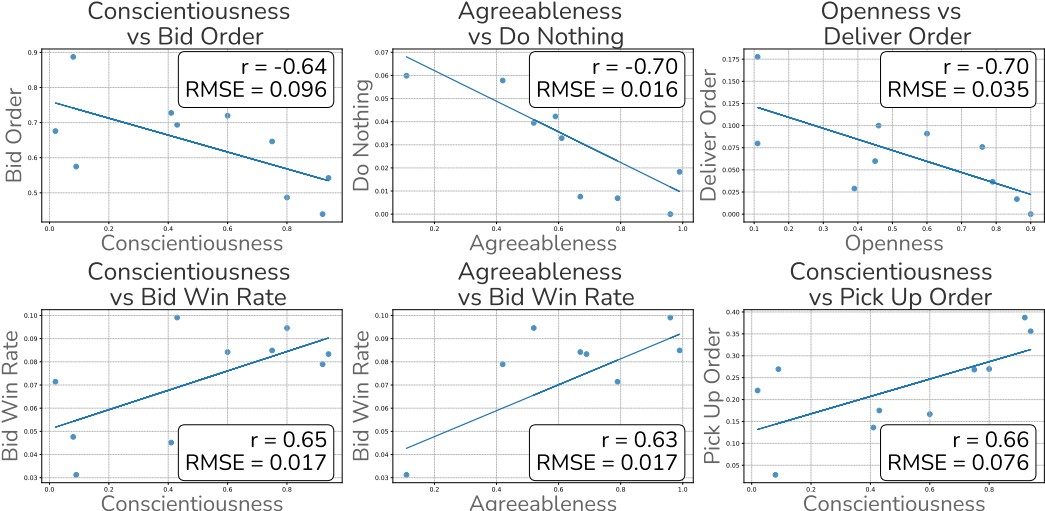

Figure 21: **Linear Regression Reveals Strong Correlations between Behaviors and Persona Traits.** The results demonstrate that agent behaviors are tightly aligned with their corresponding persona attributes, highlighting the effectiveness of trait-driven behavior modeling.

```
Action: Bidding
Reasoning:
Case1="""Since I don't have any money yet and there are good orders near me, I
    will bid 142.0 for Order(customer_position=Vector(x=-18300.0, y=-1700.0),
    store_position=Vector(x=-21700.0, y=1700.0)) since it's close to my current
     position and has a good profit margin.
"""

Case2="""
Since I have no money and no active orders, I should bid on Order(max_sale_price
    =146.5775036765818, min_sale_price=114.8202263455568) which starts near my
    location at Vector(x=38300.0, y=-1700.0) to maximize efficiency.
"""

Case3="""
Since I have no orders and no money, I'll bid for Order 2 which delivers between
     -58300,-1700 and -58300,1700 since it's closest to my current location and
     has a short estimated time of 68 seconds.
"""

Case4="""
As I have 0 dollars and see other delivery men bidding 130-140 for the first
    order, I should bid 143 dollars for Order 9 which has high max price and is
     close to store position to start earning money.
"""

Case5="""
Since I already bid an order but lost to competitor who bid 130, I need to bid a
     higher price of 135 for Order(customer_position=Vector(x=-18300.0, y
    =-1700.0), store_position=Vector(x=1700.0, y=-21700.0)) to increase my
    chances of winning.
"""

Action: Change Speed
Reasoning:
Case1="""
Since I have no orders to deliver and there are no new orders to bid on, I
    should change my walking speed to 200 cm/s to conserve energy while still
    maintaining reasonable mobility.
"""

Case2="""
I have an order to pick up at Vector(x=-21700.0, y=1700.0), so I need to move
    through Vector(x=21700.0, y=1700.0) to get there, but I've been moving back
```

```
        and forth inefficiently, so I should change my walking speed to 200 cm/s
        to conserve energy while maintaining progress.
"""
Action:Pick Up Order
Reasoning:
Case1="""
Since I already bid for two orders and have high competition, I should move
        towards Vector(x=18300.0, y=1700.0) to get closer to my assigned order's
        store location at Vector(x=38300.0, y=-1700.0).
"""

Case2="""
I need to go to Vector(x=-18300.0, y=-1700.0) to compete with other delivery men
        for orders near there since there are multiple orders in that area.
"""

Case3="""
I am at the edge of the map and low on money, so I will move towards Vector(x
        =38300.0, y=1700.0) to be closer to the center where orders are
        concentrated.
"""

Case4="""
I need to move towards Vector(x=38300.0, y=1700.0) to pick up my order at the
        store since I have already won the bid for the delivery to Vector(x
        =21700.0, y=-21700.0)
"""

Case5="""
I have two orders to pick up and I'm closer to the first store at (-58300, 1700)
        , so I'll move towards Vector(-41700.0, 1700.0) to pick up the first order.
"""

Action:Deliver Order
Reasoning:
Case1="""
I will go to Vector(x=38300.0, y=1700.0) since it's on the path to the store of
        my current order at Vector(x=-21700.0, y=1700.0).
"""

Case2="""
I need to deliver order at Vector(x=-58300.0, y=-1700.0) since I have already
        picked it up and I'm at the store position, so I will move to Vector(x
        =-58300.0, y=-1700.0).
"""

Case3="""
I have an order to deliver to Vector(x=-1700.0, y=-1700.0) and I'm at the store
        location, so I'll move to Vector(x=21700.0, y=-1700.0) which is on the
        shortest path to the customer.
"""

Case4="""
I should move to Vector(x=-18300.0, y=-1700.0) to deliver the order since it's
        closest and will let me earn money quickly.
"""

Action:Buy Beverage
Reasoning:
Case1="""
I am at the edge of the map with low energy and multiple nearby orders, so I
        should buy a beverage to recover energy before taking on more deliveries.
"""

Case2="""
My energy is getting low at 80 and I have a long delivery ahead, so I should buy
         a beverage to recover energy before heading to the store location.
"""

Case3="""
I have a picked up order but my energy is getting low, so I should buy a
        beverage first to ensure I can complete the delivery safely.
"""

Action: Buy Bike
Reasoning:
Case1="""
```

```
        I have several orders at my current location but need to pick them up from
            distant stores, so I should buy a bike first to increase efficiency since I
             have enough money
    """

    Action: Shared Order
    Reasoning:
    Case1="""
    DeliveryMan 1:I need to deliver my current orders but have low energy, so I
            should share an order to reduce my workload and conserve energy.

    DeliveryMan 4: Since DeliveryMan 1 has opened a shared order with a meeting
            point at Vector(x=18300.0, y=-1700.0), I should move to Vector(x=18300.0, y
            =-1700.0) to accept and join the shared delivery.

    DeliveryMan 1: Since I am at the meeting point Vector(x=1700.0, y=-1700.0) and
            DeliveryMan 4 wants to join my shared order, I should wait here to meet him
             .

    DeliveryMan 4: Since I am at the meeting point and have low energy but not
            enough money for beverage, I should wait here to get my share from the
            shared orders to earn money first.

    DeliveryMan 1: I don't have enough money or energy to take new orders, and my
            first shared order has been waiting too long, so I should cancel the shared
             order Order(max_sale_price=142.4723211683914, min_sale_price
            =114.17753296249396, customer_position=Vector(x=-18300.0, y=-1700.0),
            store_position=Vector(x=1700.0, y=-21700.0)) and do it myself.
    """
```

Figure 22: **Example Chain of Thought from** `Claude-3.5-Sonnet` **on a Delivery Task.**

# D Related Work

**Minecraft and Web-based Agents**   Recent advancements have led to the development of intelligent agents utilizing open-ended platforms such as Minecraft and the Web. Minecraft offers a procedurally generated, sandbox environment that supports long-horizon planning, multi-agent collaboration, and complex tool use. A series of benchmark efforts have emerged in this setting: Voyager [47] enables lifelong learning and autonomous skill acquisition through LLM-powered exploration; Team-Craft [32] evaluates collaborative agent behaviors grounded in multimodal perception; Odyssey [29] incorporates a comprehensive skill library with a fine-tuned LLaMA-3 model to enhance agent capabilities; and Optimus-2 [28] demonstrates hierarchical control using MLLMs for complex mission execution.

Parallel research has extended to web-based agents, where the environment entails realistic interaction patterns, long-term planning, and diverse user-facing tasks. WebVoyager [18], for example, builds an end-to-end system using multimodal models to support browsing, form-filling, and question answering. Complementary frameworks such as BrowserGym [7], GPT-4V Web Agent [60], Skill-Weaver [61], and Real-World WebAgent [15] further push the boundary by enabling self-improvement, long-context reasoning, and programmatic action synthesis.

Building on these foundational efforts, we introduce SIMWORLD and the Delivery Benchmark to explore a new frontier in agent research: LLM-driven multi-agent behavior in large-scale, dynamic, urban environments. Unlike prior setups, our benchmark places agents in a photorealistic, procedurally generated city and tasks them with fulfilling complex delivery jobs over extended time horizons. To meet the demands of realism and open-ended interaction, we incorporate a hierarchical control framework, persistent memory modules, in-context reasoning feedback, and personality-driven behavioral diversity. Together, these components enable immersive and coordinated agent performance, offering a robust testbed for studying language-conditioned reasoning, cooperation, and adaptation in real-world inspired settings.

**Physical and Social Simulator for AI Agents**   Simulations have played a crucial role in constructing environments for training and evaluating autonomous agents. Text-based simulators often emphasize social scenarios, such as human interaction [55], daily activities [36], and relational polarization [38]. Popular embodied simulators support a broader range of applications, particularly in embodied AI research and 2D/3D scene synthesis [27]. However, most embodied simulators remain constrained to either indoor household environments (e.g., AI2-THOR [23], Habitat [39], iGibson [24]) or outdoor driving scenarios (e.g., CARLA [10], MetaDrive [26]) or natural scenes (e.g., AirSim [43]). Most of these simulators [10, 39, 24, 43, 48, 13] often rely on a limited number of manually crafted scenes, which hinders scalability and diversity. Some platforms, such as MetaUrban [52], MetaDrive [26] and AI2-THOR [23], introduce rule-based procedural generation to alleviate this issue. Nonetheless, existing embodied simulators typically lack support for dynamic multi-agent interactions in complex, outdoor environments.

Recent advancements have introduced large-scale, language-driven social simulators capable of modeling complex societal behaviors. OASIS [55] simulates up to one million LLM-powered agents interacting on social media platforms, capturing phenomena such as information diffusion, echo chambers, and polarization. Casevo [22] integrates chain-of-thought reasoning, retrieval-augmented generation, and customizable memory mechanisms to simulate intricate social phenomena and decision-making processes. MineLand [57] offers a multi-agent Minecraft environment where agents, driven by physiological needs and limited multimodal perception, engage in collective behaviors, fostering ecological and detailed simulations. Project SID [2] further advances this landscape by deploying over 1,000 autonomous AI agents within a Minecraft environment to explore the emergence of AI civilizations. Utilizing the PIANO (Parallel Information Aggregation via Neural Orchestration) architecture, agents interact in real-time, developing specialized roles, adhering to and modifying collective rules, and engaging in cultural and religious transmission. These simulations demonstrate agents' capabilities in forming complex social structures, economies, and governance systems, providing insights into large-scale societal simulations and agentic organizational intelligence.

In summary, while prior simulators have demonstrated success in specific domains such as indoor navigation or autonomous driving, none have been explicitly designed to support dynamic, multi-agent interactions in outdoor, city-scale environments. SIMWORLD fills this gap by offering a scalable

platform that enables multi-agent collaboration and competition, language-grounded interactions, and procedural generation of urban scenes—all essential for benchmarking advanced embodied agents.

**Benchmarks for Foundation Model Agents**   With the rise of foundation models, a wide range of agent frameworks have been developed that leverage large language models (LLMs)[6, 65] or multimodal large language models (MLLMs)[54, 31, 35] to perform complex tasks. Numerous benchmarks have emerged to evaluate LLM-based agents' capabilities across domains such as decision-making[30], tool use[40], scientific reasoning[51], and software development[51, 21]. For MLLM-based agents, common benchmarks often involve embodied tasks such as vision-language navigation (VLN)[45], robotic manipulation[59, 33, 62], and interactive household activities[25, 5]. And some benchmarks focus on playing games, especially Minecraft[57]. However, most of these benchmarks present relatively short-horizon or goal-directed tasks with limited action spaces—often consisting of just a few discrete options. Agents are typically instructed to win a game[6] or complete a predefined task[54, 31, 35], which oversimplifies real-world reasoning and coordination challenges. Moreover, most settings ignore the complexity introduced by multiple agents operating in the same environment[54, 31, 30, 40, 8], where cooperation, competition, negotiation, and resource contention are vital aspects of intelligent behavior.

In contrast to benchmarks that focus on short-horizon, single-agent tasks with limited action spaces, our proposed Delivery Benchmark introduces a dynamic, multi-agent urban simulation where agents must compete, cooperate, and plan over extended time horizons. The open nature of the environment and the need for real-time resource management provide a more realistic and rigorous test bed to evaluate reasoning, coordination, and adaptation in agents based on LLM and MLLM.

# E Conclusion

In this work, we introduce SIMWORLD, a next-generation simulator designed to bridge the gap between high-level cognitive reasoning and low-level embodied execution for LLM/VLM-powered agents. SIMWORLD uniquely combines realistic physical dynamics, socially grounded interactions, and language-controllable procedural generation to create open-ended, urban-scale environments. Through two comprehensive case studies—urban physical navigation and multi-agent social delivery economy—we demonstrate the platform's capability to support complex physical and social reasoning, long-horizon planning, and strategic interaction under diverse environmental conditions.

Our evaluations reveal that while frontier LLMs exhibit promising planning and interaction abilities, they also manifest limitations in perceptual grounding, rule-following, and consistent decision-making. Importantly, SIMWORLD enables a fine-grained analysis of agent behavior under varying resource conditions, competitive pressures, and personality traits, offering new insights into the nature of agent adaptability and emergent strategies.

We believe SIMWORLD lays the foundation for a more holistic and rigorous development of embodied intelligence. By supporting scalable simulation, rich agent interfaces, and systematic evaluation, it opens up new research opportunities across AI, robotics, social simulation, and multi-agent systems. We release SIMWORLD as an open platform and invite the community to use, extend, and build upon it toward the goal of general-purpose intelligent agents capable of thriving in real-world physical and social environments.

# F Broader Impacts

SimWorld aims to establish a new paradigm for simulating realistic, interactive, and open-ended environments that support large language model (LLM)-driven agents. Its design bridges the gap between traditional simulators and the growing need for multimodal, embodied reasoning and decision-making capabilities. The broader impacts of SimWorld span multiple domains:

**Advancing AI Safety and Alignment.** SimWorld offers a controlled yet complex testbed for studying agent behavior, ethics, and alignment in real-world inspired scenarios. By supporting language-based control and social decision-making, it allows researchers to systematically evaluate how agents interpret ambiguous instructions, negotiate trade-offs, and adhere to human values — key to advancing safe and aligned AI systems.

**Democratizing Embodied AI Research.** Built on top of Unreal Engine and designed with modular extensibility, SimWorld lowers the entry barrier for researchers and educators working on embodied agents, human-AI interaction, and reinforcement learning. It enables experimentation with low-cost models while supporting high-fidelity simulation, which makes embodied AI research more accessible and inclusive.

**Benchmarking and Transparency.** SimWorld includes standardized multi-agent tasks and benchmark protocols, allowing fair comparison across models and strategies. This promotes reproducibility and transparency in a field where rapid progress has often outpaced rigorous evaluation.

**Real-World Applicability.** By simulating city-scale logistics, physical interaction, and social coordination tasks, SimWorld directly contributes to domains such as autonomous driving, service robotics, and digital urban planning. These applications can have significant societal impact, improving safety, accessibility, and efficiency in public infrastructure.

**Risks and Mitigations.** As with any simulation platform, there is a risk of overfitting models to simplified environments. We mitigate this by supporting real-world data integration and promoting diverse task settings. Moreover, the misuse of SimWorld in developing deceptive or manipulative agents is a concern; we encourage the community to adopt and share responsible usage practices, and we plan to release only non-malicious agent baselines by default.

In summary, SimWorld provides a generalizable, ethical, and forward-looking testbed that supports the next generation of interactive AI systems. Its broader impact lies in enabling the research community to move beyond static benchmarks toward dynamic, real-world–relevant evaluation and development.

