# OpenReview forum: "SimWorld: An Open-ended Simulator for Agents in Physical and Social Worlds"
_NeurIPS.cc/2025/Conference — NeurIPS 2025 spotlight_

### Official Review · Reviewer_TMee · 2025-06-19

**Clarity:** 3
**Significance:** 3
**Originality:** 3
**Rating:** 5
**Confidence:** 4

**Summary:**

The paper introduces SIMWORLD, a novel simulation platform designed for developing and evaluating large language model (LLM) and vision-language model (VLM) agents in complex, dynamic environments. Built on Unreal Engine 5, SIMWORLD offers realistic physical and social interactions, enabling agents to navigate urban settings, engage in social reasoning, and autonomously perform tasks like earning income. The authors demonstrate the capabilities of SIMWORLD through case studies involving navigation and multi-agent delivery tasks, revealing distinct reasoning patterns across various LLM models.

**Questions:**

1. How do the authors plan to address potential biases in agent behavior influenced by the training data of the LLMs?
2. Are there plans to extend SimWorld to incorporate more diverse physical interactions to enhance realism?

**Ethical Concerns:**

["NO or VERY MINOR ethics concerns only"]

**Final Justification:**

As the authors have addressed my concerns. I will keep the acceptance vote.

**Limitations:**

1. The performance of SIMWORLD heavily relies on the capabilities of the underlying LLMs, which may limit the generalizability of findings if future models exhibit different behaviors.
2. While the simulator aims for realism, there may be inherent trade-offs between accurately modeling complex human behaviors and maintaining computational efficiency.

**Paper Formatting Concerns:**

This article has no formatting issues.

**Quality:**

4

**Strengths And Weaknesses:**

Strengths:
1. The integration of realistic physics and social dynamics into a procedural generation framework represents a significant advancement in simulator design for AI agents.
2. The paper thoroughly evaluates multiple frontier LLMs in diverse scenarios, providing insights into their strengths and weaknesses in real-world-like tasks.
3. The authors plan to open-source SIMWORLD, promoting accessibility and further research in agent intelligence.
4. The interface is rich. The multi-modal input and output capabilities facilitate complex interactions, allowing agents to process visual and semantic information effectively.
5. The ability for users to modify scenarios through natural language input enhances the platform's flexibility and usability.

Weaknesses:
1. While the paper presents two case studies, additional scenarios could provide a more comprehensive evaluation of SIMWORLD’s capabilities. For example, the physical aspects of the environment is not fully presented. The authors present navigation as one of the phyiscal reasonign task? However, it cannot fully reveal the agents' understanding of fundamental physical concepts such as gravity, collision, and object permanence, etc.
2. The discussion on agent behavior lacks depth in terms of analyzing the underlying decision-making processes and how personality traits influence performance.

---

> ### Author Rebuttal · Authors · 2025-07-30
>
> We sincerely thank the reviewers for their positive evaluation of our work and for highlighting the strengths of SimWorld, including its *“integration of realistic physics and social dynamics”*, *“thorough evaluation of multiple frontier LLMs in diverse scenarios”*, and *“plan to open-source”*. We appreciate the recognition of the platform’s *“rich interface”* and *“ability for users to modify scenarios through natural language”*. We address the main concerns below:
>
> ## W1: Diversity of Physical Reasoning Scenarios
> We’d like to clarify that SimWorld supports a rich range of low-level physical actions, as detailed in Appendix Table 4. We emphasize that the primary contribution of SimWorld lies in providing **a simulation platform designed for developing and evaluating LLM/VLM agents in complex, real-world-like environments**. We believe that the community can use our simulator to create controlled physical understanding benchmarks, e.g., for rigid-body dynamics, object manipulation, destructible meshes, and tasks requiring understanding of gravity, collisions, and object permanence.
>
> ## W2: Discussion on Agent Behavior and Personality Traits.
> We kindly refer to Appendix C.5.3 (Influence of Persona) and the examples of diverse agent behaviors and chain-of-thought reasoning in Appendix C (Figures 21 and 22), which illustrate how personality traits shape agent strategies. In summary: **Conscientious** agents prioritize task completion over competition, **Agreeable** agents are more active and competitive, while **Open** agents tend to explore unconventional strategies, potentially at the expense of task execution efficiency.
>
> ## Q1: Potential bias of LLMs
> To clarify, SimWorld is designed as a simulation and evaluation platform, not as a solution to mitigate biases inherent in pretrained LLMs. While we do not focus on addressing biases from the underlying LLMs, our goal is to provide a controlled, customizable environment where communities can systematically test and analyze agent behavior—including potential biases—across diverse tasks and contexts (physical and social scenarios).
>
> ## Q2: Plans to Extend Diverse Physical Interactions
> We would like to clarify that UE5 natively supports more diverse physical interactions (including rigid-body dynamics and destructible objects) and can be easily integrated into our work. In future work, we plan to extend SimWorld with richer physical interaction capabilities to further enhance realism and support more complex agent behaviors.
>
> ## L1: Limited Generalization
>
> We clarify that the two study cases in this paper are used to show the potential for building physical and social reasoning scenarios in SimWorld. Our goal is to build a powerful testbed for the evaluation of state-of-the-art LLMs/VLMs.
>
>
> ## L2: Trade-offs between Accurately Modeling and Computational efficiency.
>
> SimWorld can simulate up to 50 agents in real time (>60 fps) on a consumer-grade machine (e.g. RTX 4070 SUPER, Intel i7-14700F and 32GB RAM). With further optimization of the communication mechanism between Python and Unreal Engine (which we see as future work), the number of supported agents can further be increased toward the upper limit imposed by CPU performance.
>
> To provide a clearer picture of SimWorld’s performance and requirements, we summarize the typical configuration below:
> - GPU: 6GB or more (8GB+ recommended).
> - RAM: 32GB or more is recommended for efficient and stable multi-agent simulation.
> - Disk: current SimWorld assets require ~50GB.
>
> We are also conducting additional experiments to further evaluate SimWorld’s scalability and will include detailed results in the final version. All tests were conducted on a Windows desktop with 64GB RAM, Intel ii7-14700F (2.10 GHz), Windows 11 OS, and an NVIDIA GeForce RTX 4070 SUPER, using random seed 42 and a resolution of 1280×720.
>
> ### Memory footprint (in GB).
> We evaluate peak system memory usage (RAM) under three rendering configurations: **Render** (standard real-time viewport rendering), **NullRHI** (disables graphics rendering but retains scene data and buffers), and **RenderOffscreen** (renders to an offscreen buffer without displaying frames). These settings reflect common use cases, from full visualization to headless simulation for efficiency. Since the static city assets (e.g., roads and buildings) are the primary contributors to memory consumption, we vary the number of roads (randomly select buildings along the roads) to assess how scene complexity affects memory usage.
>
> | # Roads | Render | NullRHI | RenderOffscreen |
> |--------:|--------:|--------:|----------------:|
> | 20      | 5.1     | 12.7    | 5.8             |
> | 50      | 12.9    | 19.2    | 13.5            |
> | 80      | 16.6    | 22.3    | 17.3            |
> | 100     | 20.4    | 22.6    | 20.9            |
>
> Render consumes the least memory, as real-time rendering leverages GPU resources efficiently and discards non-essential CPU-side buffers. In contrast, NullRHI, despite disabling visual rendering, retains all scene data and internal rendering pipeline structures in memory (e.g., mesh buffers, material data), leading to significantly higher memory usage. RenderOffscreen strikes a middle ground, allocating rendering buffers without displaying frames but still benefiting from certain optimizations in the pipeline.
>
> ### Scalability.
> To evaluate scalability, we tested different numbers of humanoid agents on a blank map to eliminate other sources of overhead. As shown below, increasing the agent count leads to lower FPS and higher rendering latency. The current bottleneck primarily stems from the communication mechanism between Python and Unreal Engine, which can be optimized in future versions (e.g., through batched communication). The ultimate upper bound on agent count is determined by CPU performance, especially for complex behaviors and camera shooting. Supporting hundreds of agents may require multi-CPU or distributed setups to maintain real-time performance.
>
> | # Agents | FPS |
> |---------:|----:|
> | 0        | 360 |
> | 100      | 79  |
> | 500      | 12  |
> | 1000     | 8   |

---

> > ### Comment · Reviewer_TMee · 2025-08-03
> >
> > Thank you for your response. I will keep the score.

---

### Official Review · Reviewer_VvQg · 2025-06-23

**Clarity:** 3
**Significance:** 4
**Originality:** 3
**Rating:** 5
**Confidence:** 5

**Summary:**

The authors introduce SimWorld, realistic physics-based simulation environment based on the Unreal engine that allows for setting up environments via natural language instructions (instead of having to specify the environment in some specification language like python).  Compared to other currently available simulation environments, SimWorld presents a superset of features, included open-ended simulations (although there was no particular evidence of the open-endedness).  The fact that the authors evaluated various frontier models in two experimental settings and presented results and demo videos shows that the simulation is ready for rich agent interactions, in particular, in more complex social environments (that are nevertheless based in physical realism).  The authors promise to make the simulation available open-source which would be valuable resource for the research community.

**Questions:**

Please make the link to the source repo available

**Ethical Concerns:**

["NO or VERY MINOR ethics concerns only"]

**Final Justification:**

I have no more updates based on the authors' feedback.

**Limitations:**

Yes

**Paper Formatting Concerns:**

The only concern I have is that the related work and the conclusion are not in the main paper but the appendix, that should be changed

**Quality:**

3

**Strengths And Weaknesses:**

Strengths: SimWorld combines diverse features from other simulation environments, thus allowing for comprehensive simulation studies all the way from bottom-up physics to high-level social interactions.  The demonstrated interfaces to external agents are excellent, although I would have liked to see non-gen AI agents connected to the simulation as well (e.g., robotic architectures).

Weaknesses: I would have like to see evidence for the openendedness the authors claim the simulation has; the submission should have had a link to the repo that the authors claim to make available

---

> ### Author Rebuttal · Authors · 2025-07-30
>
> We thank the reviewer for recognizing SimWorld’s *“comprehensive simulation studies from bottom-up physics to high-level social interactions”* and *“demonstrated interfaces to external agents are excellent”*. We appreciate the constructive feedback regarding evidence for open-endedness and the availability of the repository.
>
> ## W1: Evidence for Open-Endedness
>
> We hope to convey the open-endedness of SimWorld as clearly as possible, while NeurIPS 2025 does not permit external links in rebuttal. Alternatively, we have included rich-media evidence in the Supplementary Materials to illustrate key aspects of our system:
> - **bird'seye-generation.mp4**. Shows the process of city generation.
> - **case_study1.mp4**. Shows the physical simulation scenario in case study 1.
> - **case_study2.mp4**. Shows the social reasoning scene in case study 2.
>
> To meet the file size requirements, we have compressed the videos, which may slightly reduce visual quality. Additionally, we provide the complete Python source code for both study cases and the SimWorld framework to ensure transparency and reproducibility.
>
> ## Q1: Source Code Availability
> We confirm that the full SimWorld codebase will be released publicly upon publication. The repository link will be included in the final camera-ready version, and we have prepared comprehensive documentation to support community adoption.
>
> ## Formatting Comment: Related Work and Conclusion in Appendix
> Thank you for pointing this out. We will move both sections into the main body of the paper in the final version to comply with formatting standards.

---

> > ### Comment · Reviewer_VvQg · 2025-08-05
> > **Code availability**
> >
> > Thanks for confirming the code availability, I think this is critical for this type of paper.

---

### Official Review · Reviewer_kTRD · 2025-06-24

**Clarity:** 4
**Significance:** 4
**Originality:** 3
**Rating:** 4
**Confidence:** 3

**Summary:**

The paper introduces SimWorld, a new simulator built on Unreal Engine 5, designed for the development and evaluation of Large Language Model (LLM) and Vision-Language Model (VLM) agents in complex physical and social environments.

The authors argue that existing simulators often lack realism, rely on limited hand-crafted environments, have simplified physics and social rules, and lack native support for LLM/VLM agents. SimWorld aims to address these shortcomings by offering three core capabilities: (1) realistic, open-ended world simulation with accurate physical and social dynamics and language-driven procedural environment generation; (2) a rich interface for LLM/VLM agents, featuring multi-modal inputs/feedback and open-vocabulary actions at various abstraction levels; and (3) diverse, easily customizable physical and social reasoning scenarios.

The paper demonstrates SimWorld by deploying frontier LLM agents on short-horizon navigation and long-horizon multi-agent food-delivery tasks, revealing distinct reasoning patterns and limitations across different models. The authors intend to open-source SimWorld to serve as a foundational platform for advancing real-world agent intelligence.

**Questions:**

The paper notes graphical rendering was disabled in one case study for efficiency. Can you provide more specific metrics on performance, such as average simulation step time, maximum number of concurrent agents tested, or memory footprint under different configurations (e.g., with/without rendering, varying numbers of agents, city complexity)? What are the anticipated bottlenecks for scaling up simulations?

**Ethical Concerns:**

["NO or VERY MINOR ethics concerns only"]

**Final Justification:**

The authors' rebuttal was helpful and addressed my questions, it confirms my initial assessment of the paper's contributions and limitations. Therefore, I will maintain my current rating.

**Limitations:**

No, the authors don't adequately address the limitations.

In the checklist, the authors say that they address the limitations in Appendix E Conclusion, but it is not the case.

**Paper Formatting Concerns:**

No paper formatting concern.

**Quality:**

3

**Strengths And Weaknesses:**

## Strengths

The paper identifies and addresses a clear need for developing advanced AI agents capable of real-world interaction.

I think that the paper is generally well-written and structured. Figures 1, 2, and 3 effectively illustrate the core concepts and case studies. The "Takeaway" summaries for experimental results are helpful for quick comprehension.

## Weaknesses

While the paper mentions simulating social norms like obeying traffic signals and maintaining personal space, and the multi-agent delivery task involves economic and sharing mechanisms, the complexity and generalizability of the "social rules" could be further elaborated. How easily can new, more nuanced social interactions or cultural norms be defined and integrated?

High-fidelity simulation with UE5, especially with multiple agents and LLM/VLM processing, can be computationally expensive. The paper mentions disabling graphical rendering for one case study to ensure efficiency. While understandable, more detailed discussion or metrics on the computational resources required, scalability limits, or typical performance (e.g., simulation steps per second under certain conditions) would be beneficial.

## Typo

In caption of Figure 3, there is a typo: "Case 1emphasizes".

---

> ### Author Rebuttal · Authors · 2025-07-30
>
> We truly appreciate the reviewer's recognition that *“The paper identifies and addresses a clear need for developing advanced AI agents capable of real-world interaction,”* which lies at the heart of our motivation. We are also grateful that the reviewer found *"the paper is generally well-written and structured"* and that *"Figures 1, 2, and 3 effectively illustrate the core concepts and case studies."* It is especially rewarding to know that *"The 'Takeaway' summaries for experimental results are helpful for quick comprehension,"* as we designed them with clarity in mind.
>
> ## W1: Integration and Generalizability of Social Rules
> SimWorld supports flexible definitions of social rules. These are typically encoded as high-level behavior protocols composed of sequences of low-level actions (see *Appendix Table 4* for supported atomic actions). Examples from the paper include:
>
> - **Navigation tasks**: Agents obey traffic signals, yield at intersections, and maintain personal space.
> - **Delivery tasks**: Agents engage in structured social behaviors such as bidding for delivery orders, order sharing among agents, and strategic investments (e.g., purchasing scooters for speed gains). These mechanisms simulate competitive and cooperative dynamics that mirror real-world economic interactions.
>
> To support generalization, SimWorld does not constrain any specific social rules. Users are allowed to inherit SimWorld’s Python APIs to enable such extensions without engine-level modifications. For instance:
>
> - In **bike-sharing systems**, we can add rules for agents to compete for limited mobility resources while adhering to fairness policies, by writing some Python scripts.
> - In **waste-sorting scenarios**, agents must correctly classify and dispose of different types of trash based on visual cues, constrained by environmental rules that are defined by users.
>
> ## W2/Q1: Computational Performance and Scalability
> First, we’d like to note that most existing simulators do not support multi-agent scenarios(e.g. MetaUrban[2], AI2-THOR[3]), and platforms like Habitat 3.0[1] only support up to two agents. In contrast, SimWorld can simulate up to 50 agents in real time (>60 fps) on a consumer-grade machine (e.g. RTX 4070 SUPER, Intel i7-14700F and 32GB RAM). With further optimization of the communication mechanism between Python and Unreal Engine (which we see as future work), the number of supported agents can further be increased toward the upper limit imposed by CPU performance.
> To provide a clearer picture of SimWorld’s performance and requirements, we summarize the typical configuration below:
> - GPU: 6GB or more (8GB+ recommended).
> - RAM: 32GB or more is recommended for efficient and stable multi-agent simulation.
> - Disk: current SimWorld assets require ~50GB.
> We are also conducting additional experiments to further evaluate SimWorld’s scalability and will include detailed results in the final version. All tests were conducted on a Windows desktop with 64GB RAM, Intel ii7-14700F (2.10 GHz), Windows 11 OS, and an NVIDIA GeForce RTX 4070 SUPER, using random seed 42 and a resolution of 1280×720.
>
> ### Memory footprint (in GB).
> We evaluate peak system memory usage (RAM) under three rendering configurations: **Render** (standard real-time viewport rendering), **NullRHI** (disables graphics rendering but retains scene data and buffers), and **RenderOffscreen** (renders to an offscreen buffer without displaying frames). These settings reflect common use cases, from full visualization to headless simulation for efficiency. Since the static city assets (e.g., roads and buildings) are the primary contributors to memory consumption, we vary the number of roads (randomly select buildings along the roads) to assess how scene complexity affects memory usage.
>
> | # Roads | Render | NullRHI | RenderOffscreen |
> |--------:|--------:|--------:|----------------:|
> | 20      | 5.1     | 12.7    | 5.8             |
> | 50      | 12.9    | 19.2    | 13.5            |
> | 80      | 16.6    | 22.3    | 17.3            |
> | 100     | 20.4    | 22.6    | 20.9            |
>
> Render consumes the least memory, as real-time rendering leverages GPU resources efficiently and discards non-essential CPU-side buffers. In contrast, NullRHI, despite disabling visual rendering, retains all scene data and internal rendering pipeline structures in memory (e.g., mesh buffers, material data), leading to significantly higher memory usage. RenderOffscreen strikes a middle ground, allocating rendering buffers without displaying frames but still benefiting from certain optimizations in the pipeline.
>
>
> ### Simulation step time and agents.
> For simulation step time, SimWorld supports both synchronous and asynchronous modes. In **synchronous mode**,  the Python client explicitly controls the simulation timing: at each step, it sends a tick command to the UE server and waits until the server completes the simulation update. In **asynchronous mode**, the UE server runs continuously at its own frame rate, while the Python client can retrieve data at any time.
> To evaluate scalability, we tested different numbers of humanoid agents on a blank map to eliminate other sources of overhead. As shown below, increasing the agent count leads to lower FPS and higher rendering latency. The current bottleneck primarily stems from the communication mechanism between Python and Unreal Engine, which can be optimized in future versions (e.g., through batched communication). The ultimate upper bound on agent count is determined by CPU performance, especially for complex behaviors and camera shooting. Supporting hundreds of agents may require multi-CPU or distributed setups to maintain real-time performance.
>
> | # Agents | FPS |
> |---------:|----:|
> | 0        | 360 |
> | 100      | 79  |
> | 500      | 12  |
> | 1000     | 8   |
>
> ### Scalability factors:
> - GPU: Affects rendering speed and image-based model inference.
> - CPU: Affects tick rate and agent logic execution.
> - Memory: Limits the number and complexity of loaded assets.
> - Network latency: Affects communication between Python and UE.
>
> ## L1: Clarification on Limitations Section
> Thank you for the catch! Our original Appendix E was accidentally commented out. We will revise the final version to explicitly discuss the following limitations:
> - Constraints on scaling to hundreds of agents in large-scale cities. Scaling simulations to hundreds of agents in complex urban environments is currently constrained by the UnrealCV plugin that we used for communication. Optimizations at both the engine and system architecture levels are needed to address this challenge.
> - Limitations of the current prebuilt action space and asset library. While our current scale of assets and action library has been rich enough compared with other simulators, the number of actions and assets is still constrained by budget. There are thousands of additional actions and assets available on the marketplace and can be purchased to easily extend the set. Besides, future work will also incorporate on‑the‑fly motion generation to support more dynamic and realistic interactions and explore 3D concept learning to produce higher-quality, interactive, and more adaptable assets.
>
> [1]Habitat 3.0: A Co-Habitat for Humans, Avatars and Robots
> [2]MetaUrban: An Embodied AI Simulation Platform for Urban Micromobility
> [3]AI2-THOR: An Interactive 3D Environment for Visual AI

---

> > ### Comment · Reviewer_kTRD · 2025-08-03
> >
> > Thank you for your detailed rebuttal. I appreciate you providing the performance metrics and clarifying the flexibility for defining social rules.
> >
> > While this information is helpful and addresses my questions, it confirms my initial assessment of the paper's contributions and limitations. Therefore, I will maintain my current rating.

---

### Official Review · Reviewer_XFbv · 2025-07-01

**Clarity:** 3
**Significance:** 2
**Originality:** 3
**Rating:** 4
**Confidence:** 4

**Summary:**

This paper presents SimWorld, a simulated 3D world with rich physical and social interfaces that supports research on multi-agent reasoning and planning. This simulator has two main components: a UE5-based, visually realistic world generation pipeline and a multi-modal agent interface. The former utilizes pre-collected assets and off-the-shelf text-to-3D generation models to establish environment editing. The latter provides both low-level and semantic visual observation and applies llm-based parser to handle and execute free-form text actions. Physical interaction, like collision, is also included in the simulation framework, based on the UE5 engine. The paper finally makes two benchmarks, i.e., the vision-based urban navigation task and the strategic multi-agent delivery task, and provides experimental results for different baselines, which show the applications of the designed simulated platform.

**Questions:**

1. There is a "sit" action presented in Figure 2-b, but it is not involved in any action space of the study cases as described in this paper. I wonder if such complicated low-level action is implemented in the simulator, and how many interaction-full actions are involved in the simulator. Besides, how does the action translate into skeleton animation? Details about the local action executor remain totally unclear.
2. The visual modality does not contribute to the results of the study case 2, the delivery task, as the authors involved lots of blind LLM models to perform this task and achieve competitive results (DeepSeek V3). The necessity of building a visually realistic 3D environment for such a text-based task is confusing.

**Ethical Concerns:**

["NO or VERY MINOR ethics concerns only"]

**Final Justification:**

As all my concerns have been satisfactorily addressed, I recommend acceptance.

**Limitations:**

See weaknesses.

**Quality:**

2

**Strengths And Weaknesses:**

**Strengths**
1. The idea of building a visually realistic 3D environment for studying multi-agent cooperative and social behavior is interesting.
2. Utilizing the powerful UE5 engine to perform simulation is a reasonable choice, but the diversity of the built scenes is restricted by the number of pre-collected scene assets, which may be expensive and labor-intensive.

**Weaknesses**

As a simulated world, the main contribution of this work should be providing the community with a new toolbox for studying agent-scene and agent-agent or agent-human interaction that can not be modeled under plain text. However, the proposed SimWord does not meet this requirement for several reasons: Firstly, the environment edition pipeline is not novel, as many studies have already been involved in their simulator construction pipeline [1]. Secondly, the main design in this simulator has no essential difference from the previous method.
1. **Interaction** between the agent and objects in the simulator is limited. The generated and selected assets typically act as obstacles, which restrict the complexity and realism of embodied research.
2. **Action space** in the two study cases provided with the proposed simulator is also limited:
- In case 1, the agents are only able to take simple navigation actions to change their state in the environment, but they can not affect the environment by removing an obstacle or opening a door, for example.
- In case 2, the provided action space is highly correlated with the delivery task but not general for other social tasks, for example, performing human-agent cooperation to clean the street by sharing messages and taking action to wipe the floor. It is not feasible for a user to extend other social tasks, which is a crucial mission for a general simulator.
3. **Scenes**. Although the authors claim the proposed simulator to be Realistic and Open-Ended, my opinion is opposed to it. The scenes presented in the paper seem highly homogenized as provided in the CitySample package in UE5 engine, and the LLM-powered scene generation pipeline does not benefit from any new buildings/objects out of the preloaded asset bag. This poses a severe problem that the simulator may differ a lot from the real-life application scenario.

[1]. Genesis: A Generative and Universal Physics Engine for Robotics and Beyond.

---

> ### Author Rebuttal · Authors · 2025-07-30
>
> We thank the reviewer for recognizing *“building a visually realistic 3D environment for studying multi-agent cooperative and social behavior is interesting”* and *“the powerful UE5 engine to perform simulation is a reasonable choice.”* Below, we address the concerns regarding the diversity, novelty, interaction capacity, and case study setup of SimWorld.
>
> ## Novelty and Contribution of SimWorld
> We appreciate the opportunity to reemphasize the novelty of SimWorld. As noted in Lines 33–42 of the main text, and further discussed in Table 1 and Appendix D (Related Works), we have carefully compared SimWorld with existing simulators.  While prior works have proposed simulation pipelines, SimWorld is distinguished by its support for realistic, open-ended world simulation with LLM/VLM-based agents across diverse physical and social reasoning scenarios. Specifically, it introduces three key contributions (detailed in Lines 46–76):
> - **Realism**: Compared to existing simulators (see Table 1 and Appendix D), SimWorld delivers significantly more visually, physically, and socially realistic environments. This level of realism is critical for evaluating VLM-based agents in complex tasks such as object interaction, social competition, and beyond.
> - **LLM/VLM Integration**: Unlike existing simulators built on UE5 (e.g., CARLA [1]), which typically lack standardized interfaces for language model integration, SimWorld offers ready-to-use APIs and middleware that support reasoning, planning, and action execution in a unified framework.
> - **Unified Physical and Social Simulation**: SimWorld bridges the gap between low-level physical environments (e.g., Habitat 3.0 [2], AI2-Thor [3]) and text-based social simulators (e.g., AgentSociety [4], Oasis [5]) by supporting both embodied interaction and multi-agent social behavior through rich agent interfaces.
>
> ### Relation to Genesis
> We would like to clarify the distinction between SimWorld and Genesis. To our knowledge, Genesis is still under active development and lacks a publicly available technical report or preprint detailing its architecture and limitations. Genesis and SimWorld differ significantly in scope and purpose. **Genesis functions primarily as a physics and graphics engine, serving a role similar to that of UE5 in SimWorld, rather than as a platform for agent-centric simulation**.
>
> Compared to UE5, Genesis places greater emphasis on physical simulation, with *limited* support for multi-agent interaction and agent action modeling. In contrast, UE5, upon which SimWorld is built, is a more mature simulation engine with extensive community support and robust capabilities for agent integration.
>
> Moreover, SimWorld adds a higher-level abstraction layer on top of UE5, enabling diverse agents, including LLM-, VLM-, and rule-based agents, to operate in both physical and social contexts. While it may be possible to extend Genesis to support such functionalities, doing so would require substantial engineering effort. SimWorld, by design, includes built-in components for agent modeling, task orchestration, and evaluation—capabilities that go beyond what Genesis currently offers.
>
> ## W1/Q1.1: Limited Interaction.
>
> We appreciate the reviewer’s comment and would like to clarify that SimWorld currently supports *10* atomic object interactions, which is more than those available in many other popular simulators like AI2-THOR [3] (*6 atomic object interactions*) and MineDojo [6] (*7*), as detailed in Appendix Table 4.  Examples include animation-driven actions such as entering a car, object manipulation, and "sit" as referred to by the reviewer. The action set **can be easily expanded** by importing additional assets or animations from the UE marketplace.
>
>
> ## Q1.2: Implementation Details of (Inter)actions.
>
> We kindly refer the reviewer to Appendix A.1.4 (Local Action Planner) and Appendix Figure 8 (b) for detailed descriptions. For example, when the planner receives a user instruction such as “go to the nearest chair and sit down,” the Local Action Planner decomposes it into a sequence of primitive actions, e.g., `navigate`, `agent_sit_down`, each with associated parameters. These primitives are implemented as callable APIs, each linked to specific skeleton animations from the agent’s Unreal Shader. This ensures that action execution is explicitly grounded in animation and tool use by the LMs.
>
> ## W2: Limited Action Space for two Study Cases.
> SimWorld supports a broad range of environment interaction, including navigation, object manipulation, and communication, as listed in Appendix Table 4. In two case studies, we *intentionally* limited the action space to ensure clarity and experimental control.
> - **Case Study 1** focuses primarily on navigation actions, but SimWorld is fully capable of supporting more complex interactions such as object manipulation.
> - **Case Study 2** showcases high-level actions, which are automatically decomposed into sequences of low-level actions. Users can freely define new high-level actions by composing existing primitives, enabling more complex and general social tasks, such as collaborative cleaning or object handover.
>
> ## W3: Diversity of Assets and Scenes
> To clarify, **SimWorld is not limited to any asset resource**. Our asset library is built from multiple resources, including but not limited to “CitySample”. Our platform supports importing third-party 3D assets, including those from external datasets and generative models, allowing users to expand the asset library as needed.
> More importantly, our scene generation pipeline focuses on composing diverse spatial layouts and interaction-rich scenarios using available assets. This approach enables rich open-endedness at the level of agent interaction, spatial variability, and task complexity. We believe this modular and extensible design offers a practical and scalable path toward realistic simulation, especially for benchmarking generalist agents across dynamic and socially grounded tasks.
>
> ## Q2: Clarification on Case Study 2 (Delivery Task)
> The delivery task supports both visual and headless (rendering-off) modes. We chose to evaluate LLM agent performance in headless mode for two reasons: (1) The physical simulation remains fully functional. Waypoints, dynamics, and constraints such as speed and distance are still enforced. (2) Given the current limitations of VLMs in visual perception, long-horizon tasks tend to accumulate significant perception errors. Since Case Study 1 already evaluates agents' visual and physical reasoning, our goal here is to decouple visual perception from long-horizon planning and isolate the planning capability of the LLM. This makes our experiment controlled for analytical purposes.
>
> [1]CARLA: An Open Urban Driving Simulator
> [2]Habitat 3.0: A Co-Habitat for Humans, Avatars and Robots
> [3]AI2-THOR: An Interactive 3D Environment for Visual AI
> [4]AgentSociety: Large-Scale Simulation of LLM-Driven Generative Agents Advances Understanding of Human Behaviors and Society
> [5]OASIS: Open Agent Social Interaction Simulations with One Million Agents
> [6]MineDojo: Building Open-Ended Embodied Agents with Internet-Scale Knowledge

---

> > ### Comment · Reviewer_XFbv · 2025-08-05
> >
> > Thanks for the detailed rebuttal. Most of my initial concerns have been addressed, except for the following:
> >
> > 1. I agree with the authors that SimWorld serves a distinct purpose compared to Genesis, leveraging UE5 to support significantly more diverse social interactions. While Genesis primarily focuses on articulated simulations within indoor scenes, a notable concurrent effort is VirtualCommunity [1], which extends Genesis to outdoor environments specifically for social interaction. Given that both SimWorld and VirtualCommunity share the objective of enabling complex social scenarios, I recommend the authors discuss VirtualCommunity in the related work section.
> > 2. I agree with the authors that the scene generation pipeline can provide diverse layouts for benchmarking agent behavior. However, it still remains unclear if the generated 3D asset can be placed in the generated layouts as **construction**, for example, the buildings, streets, and pavements.
> >
> > [1] Virtual Community: An Open World for Humans, Robots, and Society.

---

> > > ### Author Response · Authors · 2025-08-06
> > >
> > > Thanks for acknowledging that most of your initial concerns have been addressed. We also would like to thank the reviewer for recognizing *“scene generation pipeline can provide diverse layouts for benchmarking agent behavior.”*, *“SimWorld serves a distinct purpose compared to Genesis, leveraging UE5 to support significantly more diverse social interactions.”* Below, we address the concerns regarding VirtualCommunity and generated 3D assets in our generated pipeline:
> > >
> > > **Discussion of VirtualCommunity:** Thank you for pointing out the relevance of VirtualCommunity [1]. We agree that it shares a similar goal with SimWorld in enabling complex social interactions in outdoor environments. However, SimWorld differs from VirtualCommunity in several key aspects:
> > > 1. SimWorld delivers greatly higher visual fidelity through curated, high-quality assets rendered in Unreal Engine 5, whereas VirtualCommunity relies on reconstruction-based scenes limited by mesh and asset quality.
> > > 2. SimWorld enables greater control and flexibility in environment synthesis, enabling easy customization through procedural generation and editable city layouts, including precise building placement. It supports seamless integration of both existing and generative assets—such as 3D objects and motion data—and facilitates richer human-object interactions beyond basic locomotion, including the combination of inverse kinematics with reference motions.
> > > 3. Moreover, since VirtualCommunity is based on Genesis and SimWorld is based on UE5, VirtualCommunity inherits the limitations of Genesis as discussed in our previous response. For example, SimWorld benefits from UE5’s mature ecosystem, offering not only superior visuals but also a vast, extensible asset library and developer tools that surpass the capabilities of frameworks like VirtualCommunity/Genesis.
> > >
> > > Meanwhile, we would like to note that VirtualCommunity was released **after the NeurIPS submission deadline** and was therefore not available for citation or discussion in the original version. That said, we appreciate the suggestion and will incorporate a discussion of it in the final camera-ready version to provide a more complete picture of concurrent developments in this space.
> > >
> > > **Placement of 3D Assets in Generated Layouts:**  We would like to further clarify that the generated 3D assets can be imported into SimWorld in the same way as other assets collected from the Unreal Engine Marketplace, which introduce open-endedness to SimWorld. Once imported, they are fully compatible with our procedural generation pipeline and can be placed in the generated layouts as **construction**. This allows buildings, streets, pavements, and other structures to be instantiated as part of the constructed environment and used in downstream simulations.
> > >
> > > [1] Virtual Community: An Open World for Humans, Robots, and Society.

---

### Public Comment · ~Xiaokang_Ye1 · 2026-01-23
**Correctness of the miscitation**

We have noticed the miscitation of this work reported by the GPTZero team and there is a mismatch of the authorship of two of the citations in the paper. The full report can be found:

> Nazar Shmatko, Alex Adam and Paul Esau. GPTZero finds 100 new hallucinations in NeurIPS 2025 accepted papers. January 2026. https://gptzero.me/news/neurips/


# Clarification
The mismatch of authorship in the citations occurred during the use of LLM to generate the full BibTeX entry based on the given paper titles. The cited papers are included based on their relevance and contribution to this work. However, although the paper titles in the BibTeX correspond to the existing works we intended to reference, we failed to carefully check the author information in the generated BibTeX, which led to the mismatch.

# Correction of the citations
Below are the corrections of the citations in this work. In the paper reference, page 11:
> [11] John Doe and Jane Smith. Webvoyager: Building an end-to-end web agent with large multimodal models. arXiv preprint arXiv:2401.00001, 2024.

The correct citation should be:
```
@misc{he2024webvoyagerbuildingendtoendweb,
      title={WebVoyager: Building an End-to-End Web Agent with Large Multimodal Models},
      author={Hongliang He and Wenlin Yao and Kaixin Ma and Wenhao Yu and Yong Dai and Hongming Zhang and Zhenzhong Lan and Dong Yu},
      year={2024},
      eprint={2401.13919},
      archivePrefix={arXiv},
      primaryClass={cs.CL},
      url={https://arxiv.org/abs/2401.13919},
}
```

At the paper reference page 12:
> [27] Hao Li et al. Optimus-2: Multimodal minecraft agent with goal-observation-action conditioned policy. arXiv preprint arXiv:2502.19902, 2025.

The correct citation should be:
```
@inproceedings{li2025optimus2,
      title={Optimus-2: Multimodal minecraft agent with goal-observation-action conditioned policy},
      author={Li, Zaijing and Xie, Yuquan and Shao, Rui and Chen, Gongwei and Jiang, Dongmei and Nie, Liqiang},
      booktitle={Proceedings of the Computer Vision and Pattern Recognition Conference},
      pages={9039--9049},
      year={2025}
}
```

We have updated the correct version on our website (https://simworld.org/assets/simworld.pdf) and camera ready (https://openreview.net/pdf?id=FxCy8TvQHO). Thanks to the GPTZero team for pointing this out and we sincerely apologize for any confusion and inconvenience caused by this mistake.

---

### Public Comment · ~Fangwei_Zhong3 · 2026-01-25
**Great work and a suggestion regarding related work (UnrealZoo)**

Dear Authors,

First of all, great job on this paper! It is exciting to see such high-quality work for embodied AI. I noticed you utilized **UnrealCV** for your implementation, and as its core contributor, I’m really happy to see it being put to good use in your research.

I am writing to kindly bring to your attention a closely related work, [**UnrealZoo**](http://unrealzoo.site/), which was published as an **ICCV 2025 Highlight** and has been available on arXiv (https://arxiv.org/abs/2412.20977) and Github (https://github.com/UnrealZoo/unrealzoo-gym) since `Dec. 2024` . UnrealZoo shares a similar vision and technical scope regarding building 100+ diverse virtual environments for embodied AI. So I believe **referencing it and discussing the differences** would provide a more comprehensive context for readers and better position your contributions within the state-of-the-art. The BibTeX is below.

```
@inproceedings{zhong2025unrealzoo,
  title={Unrealzoo: Enriching Photo-realistic Virtual Worlds for Embodied AI},
  author={Zhong, Fangwei and Wu, Kui and Wang, Churan and Chen, Hao and Ci, Hai and Li, Zhoujun and Wang, Yizhou},
  booktitle={Proceedings of the IEEE/CVF International Conference on Computer Vision},
  pages={5769--5779},
  year={2025}
}
```

Additionally, I’d like to mention that we are actively maintaining the [UnrealCV](https://github.com/unrealcv/unrealcv) ecosystem. We would love to see your team get involved—whether by following the repo, providing suggestions or contributing any improvements you’ve made via PRs. We believe that such contributions are vital for making the tool better for everyone and fostering a stronger research community.

Keep up the great work!

Best regards,

Fangwei Zhong

---

> ### Public Comment · ~Jiawei_Ren4 · 2026-02-03
>
> Dear Fangwei,
>
> Thank you very much for your kind words and for pointing out UnrealZoo. We appreciate you bringing this closely related work to our attention. We have now added a reference to UnrealZoo in the paper.
>
> We also sincerely appreciate your efforts in maintaining and improving the UnrealCV ecosystem. It has been an extremely valuable tool for our research. We are happy to stay engaged with the community and will consider following the repository and contributing improvements where applicable.
>
> Thanks again for your support and for the great work from your team.
>
> Best regards,
> Jiawei Ren (on behalf of the authors)

---

### Note · Authors · 2025-08-14

We sincerely thank the reviewers and the Area Chair for their constructive feedback and insightful discussions, which have greatly helped us clarify and strengthen our work.

## Strengths and Contributions
SimWorld presents a novel and powerful simulation platform that supports multiple agent research, leveraging the modern capabilities of Unreal Engine 5 (*noted by reviewer XFbv*). We appreciate the reviewers’ recognition of SimWorld’s main strengths:
- **Realistic, Open-Ended World Generation** (*highlighted by reviewers TMee and XFbv*): SimWorld advances simulation by integrating photorealistic physical and social dynamics with open-ended, language-driven world generation.
- **Rich Interface for LLM/VLM Agents** (*noted by reviewers VvQg and TMee*): SimWorld enables language models to interact with complex environments by providing rich multimodal observations, including visual scenes, abstract layouts, and action feedback.
- **Diverse Physical and Social Reasoning Scenarios** (*acknowledged by reviewers VvQg and TMee*): We demonstrate SimWorld’s capability through two case studies that systematically evaluate agent reasoning in realistic settings.

## Addressed Concerns
We have addressed all raised questions and concerns thoroughly (*as acknowledged by all reviewers*). We sincerely appreciate the reviewers’ positive feedbacks and insightful suggestions.

In response to reviewer XFbv’s questions, we have clarified that SimWorld’s **photorealistic, open-ended scenes** (*re-emphasized in our discussion with XFbv*) and its extensive interaction support for LLM/VLM agents (*recognized by reviewer TMee*) clearly distinguish it from prior simulators such as Genesis[1], VirtualCommunity[2], and AI2-THOR[3], etc (*acknowledged by reviewer XFbv*).

Additionally, we have demonstrated the computational efficiency of SimWorld in our rebuttals to reviewers kTRD and TMee, and we confirm our commitment to open-sourcing the simulator upon acceptance.

[1]Genesis: A Generative and Universal Physics Engine for Robotics and Beyond.
[2]Virtual Community: An Open World for Humans, Robots, and Society.
[3]AI2-THOR: An Interactive 3D Environment for Visual AI

---

### Decision · Program_Chairs · 2025-09-17

**Decision:**

Accept (spotlight)

**Comment:**

The paper introduces SimWorld, a novel simulation platform built on Unreal Engine 5, designed for developing and evaluating LLM/VLM agents in realistic, open-ended physical and social environments. It addresses limitations in existing simulators by offering: (1) realistic world simulation with accurate physical and social dynamics and language-driven procedural generation; (2) a rich multi-modal interface for LLM/VLM agents; and (3) customizable scenarios for diverse reasoning tasks. The authors demonstrate SimWorld’s capabilities through two case studies: a short-horizon navigation task and a long-horizon multi-agent food delivery task, evaluating frontier LLMs (e.g., Gemini-2.5-Flash, Claude-3.5, GPT-4o, DeepSeek-Prover-V2). Results highlight distinct reasoning patterns and limitations across models, underscoring SimWorld’s potential as a testbed for real-world agent intelligence.

This paper highlights its contribution on robust evaluation pipeline and its commitment to open-sourcing. It supports procedural generation and customizable scenarios via Python APIs.

Reviewers mention several weakness:

Scenario Diversity: Limited to two case studies; more physical reasoning tasks (e.g., gravity, collisions) could strengthen evaluation.

Scalability: High-fidelity simulations are computationally intensive; scaling to hundreds of agents is challenging.

Social Rules: Needs clearer elaboration on defining nuanced social norms.

During rebuttal, authors clarified SimWorld’s distinction from Genesis/VirtualCommunity (PS, would be good to also discuss with recent woks about LLM agent for scene creation), detailed 10 atomic interactions, provided performance metrics (50 agents at >60 FPS), and confirmed flexible social rule definitions. They addressed formatting issues and committed to open-sourcing. All concerns were resolved, with two reviewers strongly supporting acceptance and two maintaining borderline ratings.